# Microstructure, Micro-inclusions and Mineralogy along the EGRIP ice core - Part 1: Localisation of inclusions and deformation patterns

Nicolas Stoll[1], Jan Eichler[1], Maria Hörhold[1], Tobias Erhardt[1, 2], Camilla Jensen[2], and Ilka Weikusat[1, 3]

[1]Alfred Wegener Institute Helmholtz Centre for Polar and Marine Research, Bremerhaven, Germany
[2]Climate and Environmental Physics, Physics Institute and Oeschger Centre for Climate Change Research, University of Bern, Bern, Switzerland
[3]Department of Geosciences, Eberhard Karls University, Tübingen, Germany

**Correspondence:** Nicolas Stoll (nicolas.stoll@awi.de)

**Abstract.** Impurities deposited in polar ice allow the reconstruction of the atmospheric aerosol concentration of the past. At the same time they impact the physical properties of the ice itself such as its deformation behaviour. Impurities are thought to enhance ice deformation, but observations are ambiguous due to a shortage of comprehensive microstructural analyses. For the first time, we systematically analyse micro-inclusions in polar fast flowing ice, i.e. from the East Greenland Ice Core Project ice core drilled trough the Northeast Greenland Ice Stream. In direct relation to the inclusions we derive the crystal-preferred orientation, fabric, grain size, and microstructural features at ten depths, covering the Holocene and Late Glacial. We use optical microscopy to create microstructure maps to analyse the in situ locations of inclusions in the polycrystalline, solid ice samples. Micro-inclusions are more variable in spatial distribution than previously observed, and show various distributional patterns ranging from centimetre-thick layers to clusters and solitary particles, independent of depth. In half of all samples, micro-inclusions are more often located at or close to the grain boundaries by a slight margin (in the areas occupied by grain boundaries). Throughout all samples we find strong indications of dynamic recrystallisation, such as grain islands, bulging grains and different types of subgrain boundaries. We discuss the spatial variability of micro-inclusions, the link between spatial variability and mineralogy, and possible effects on the microstructure and deformation behaviour of the ice. Our results emphasise the need for holistic approaches in future studies, combining microstructure and impurity analysis.

## 1 Introduction

Polar ice sheets are key elements of our climate system and deep ice cores from these regions are used, but not limited, to reconstruct the paleoclimate (e.g., Lorius et al., 1985; Petit et al., 1999; Watanabe et al., 2003; EPICA Community Members, 2004; Dahl-Jensen et al., 2013) and to investigate the dynamics of ice in ice sheets (e.g., Alley, 1988; Weikusat et al., 2017a). The atmospheric aerosol concentration is partly imprinted in the snow at the surface. As the snow transforms to ice, the deposited aerosols either dissolve in the ice structure or form micro-inclusions. With time they get transported into deeper regions in the ice sheet. As ice crystals are separated by dynamically changing interfaces, i.e. grain boundaries, they undergo constant changes in response to stress and strain resulting in different crystal shapes, sizes, and orientations. Therefore ice

cores enable the gathering of information about the internal deformation of polar ice and its mechanisms, as the localised characteristics determine the large-scale deformation behaviour.

The deformation characteristics of ice strongly impact the flow of ice sheets. The quality of future projections of the ice sheets and their ice flow under changing climate conditions strongly depends on the understanding of ice dynamics on all scales. Especially the mechanisms controlling fast flowing ice stream dynamics and the relation to the slow-moving ice outside the ice streams is insufficient (Minchew et al., 2018, 2019; Stearns and van der Veen, 2019).

    To improve the understanding of large-scale ice dynamics processes in the microstructure of ice, i.e. on the millimetre-
and centimetre-scale, have to be understood as they control the strain rate. The main deformation process of ice crystals is dislocation creep, which describes the movement of dislocations in the basal plane in combination with climb (Glen and Jones, 1967; Weertman, 1973; Hobbs, 1974; Petrenko and Whitworth, 1999). The preferred orientation of crystals plays an important role for the deformation of ice, however, under certain boundary conditions, it is further impacted by other properties such as grain size (e.g., Goldsby and Kohlstedt, 1997; Cuffey et al., 2000; Kuiper et al., 2020b), crystal preferred orientation (CPO),
and also impurity content (e.g., Glen and Jones, 1967; Jones and Glen, 1969; Fisher and Koerner, 1986). The relationship between CPO, grain size, and impurity content and how they impact deformation is still under debate (Stoll et al., 2021b). Nevertheless, the impact of impurities on the microstructural deformation has been shown to play a significant role and needs further investigation.

    In this study, we define impurities as extrinsic chemical compounds found in the ice which can be separated into soluble and
insoluble impurities. They were deposited on the ice sheet and originated from a variety of atmospheric aerosols with unique transport histories (Legrand and Mayewski, 1997; Weiss et al., 2002). Soluble impurities dissolve in the lattice and originate from dissolved gases or from salts dissociated into ions (Legrand and Mayewski, 1997; Della Lunga et al., 2014). Those impurities are chemical compounds and elements of diverse origin (marine, terrestrial, biological, atmospheric). Insoluble impurities consist of lattice-incoherent phases and are rejected from the ice lattice (Ashby, 1969; Alley et al., 1986a). They
are often from terrestrial origin, such as dust, and vary in size, ranging from micrometre-sized "micro-inclusions" to large dust particles (Steffensen, 1997; Wegner et al., 2015; Simonsen et al., 2019).

    Impurities impact the physical properties of snow, firn, and ice at all depths, ranging from permittivity (Wilhelms et al., 1998) to electrical conductivity (Alley and Woods, 1996; Wolff et al., 1997), and mechanical properties (Dahl-Jensen and Gundestrup, 1987; Paterson, 1991; Weiss et al., 2002; Hörhold et al., 2012; Fujita et al., 2014; Moser et al., 2020). Especially
the impact of impurities on the deformation of ice has been investigated for decades (e.g., Jones and Glen, 1969; Petit et al., 1987; Fukazawa et al., 1998; Iliescu and Baker, 2008; Eichler et al., 2019; Stoll et al., 2021b). Impurities can have indirect effects on the deformation of ice, i.e. they affect stress accommodating mechanisms such as recrystallization. In this case, the microstructure (grain shape and size) is impacted by impurities, which control grain boundary mobility, energy, and length by e.g., dragging of grain boundaries (Alley et al., 1986a, 1989) or Zener pinning (Smith, 1948; Humphreys and Hatherly,
2004). Impurities can have a direct impact by multiplying dislocations or increasing their mobility. This happens when there are obstacles, such as micro-inclusions, in the ice matrix which produce strain localisation and thus new dislocation lines (Weertman and Weertman, 1992), or occupy lattice sites introducing protonic defects (Glen, 1968).

Thus, understanding the microstructural impurity localisation is significant regarding dielectric properties (Stillman et al., 2013), the stratigraphic integrity of impurity records in deep polar ice (Faria et al., 2014a; Ng, 2021), and deformational properties (e.g., Dahl-Jensen et al., 1997; Barnes et al., 2002a; Della Lunga et al., 2014; Eichler et al., 2017; Shigeyama et al., 2019). Ice core impurity content in ice cores is often measured by meltwater analysis such as continuous melt water analysis (CFA) (e.g., Röthlisberger et al., 2000; McConnell et al., 2002; Kaufmann et al., 2008), and/or Ion Chromatography (e.g., Cole-Dai et al., 2006; Severi et al., 2014). However the information on the in situ microstructural localisation, composition, and form of impurities is lost.

Samples from several ice cores have been analysed with methods enabling the location of impurites, such as optical microscopy (e.g., Kipfstuhl et al., 2006; Faria et al., 2010; Eichler et al., 2017), Scanning Electron Microscope (SEM) coupled with energy dispersive X-ray spectroscopy (EDS) (e.g., Cullen and Baker, 2000; Barnes et al., 2002a; Obbard and Baker, 2007), LA-ICP-MS (e.g., Reinhardt et al., 2001; Della Lunga et al., 2014; Spaulding et al., 2017; Bohleber et al., 2020) and Raman spectroscopy (e.g., Ohno et al., 2005; Sakurai et al., 2011; Ohno et al., 2014; Eichler et al., 2019). Results were diverse, and often ambiguous, and were recently summarised by Stoll et al. (2021b). A major impediment is the often small number of analysed samples and limited amounts of measured impurities. Furthermore, samples often originate from more or less arbitrary depths, allowing glimpses into specific depth regimes with highly variable boundary conditions (e.g., age, bulk impurity content). Discussed generalisations of spatial distribution and the drawn conclusions on deformation effects were seldom based on systematic analyses, i.e. along a specific section of an ice core. Furthermore, the large variety in applied methods, sample origins and analysed impurity forms (soluble, insoluble, elements, compounds) impede the drawing of reliable conclusions for polar ice in general. Stoll et al. (2021b) concluded that, among others, structured methodological approaches are needed to enhance our understanding of the role of impurities regarding the deformation of ice. One promising approach is a systematic high-resolution analysis of one deep ice core combining methods from microstructure and impurity research.

Even though impurities do play a significant role in ice deformation, the specific species or processes are not understood, also due to limitations by measurement techniques or previous measurement set-ups. Which minerals are found throughout polar ice? In which state are impurities found and does their state influence their location in respect to grain boundaries and the crystal lattice? And finally, which of these (and probably many others) properties impact the deformation rate of ice and are they vice-versa affected by deforming ice? The on-going East Greenland Ice Core Project (EGRIP) located on the Northeast Greenland Ice Stream (NEGIS) is a chance to address these questions. The EGRIP core is the first deep ice core from an ice stream and thus an unique possibility to study ice dynamics in detail as displayed in recent CPO and visual stratigraphy data (Westhoff et al., 2020).

Addressing these interdisciplinary questions is extensive work and thus presented in two companion papers. In this study we investigate the localisation of visible micro-inclusions and deformation patterns along the EGRIP ice core while Stoll et al. (2021a) analyse the mineralogy of these micro-inclusions. Here we apply optical microscopy and automated fabric analyser measurements with related statistical analysis of the microstructure on ten samples covering Holocene and Late Glacial ice. We aim to give a systematic overview of the microstructural locations of micro-inclusions and of the evolution of the microstructure

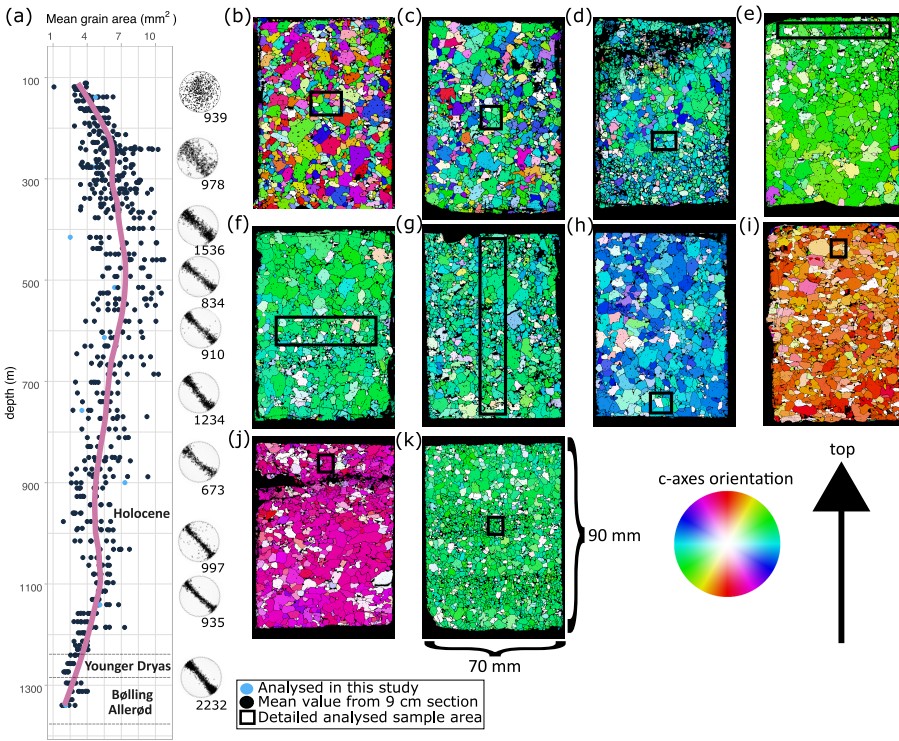

**Figure 1.** Grain size, CPO and fabric data from the upper 1340 m of the EGRIP ice core. a) 9 cm section mean grain sizes derived via FA G50. The violet line is a locally weighted regression with a smoothing parameter of 0.3. C-axis orientations of each section projected into a horizontal plane rotated towards their most likely position (Westhoff et al., 2020), the true orientation was lost during drilling. The number is c-axes per section. b)-k) Fabric data from the analysed thin sections. The colour code (legend) indicates the c-axes orientation, vertical c-axes appear white. Fabric image surfaces are not the same as in impurity maps due to sample processing and the focusing into the sample. Black areas are background corrections.

and discuss mineralogy-depended spatial patterns. Finally, we elaborate the effects of micro-inclusions on the deformation behaviour as seen in the microstructure of the EGRIP ice core.

## 2  Methods

### 2.1  The East Greenland Ice Core Project

EGRIP is an ice core drilling project located on NEGIS, the largest ice stream in Greenland which terminates in three outlet glaciers (Nioghalvfjerds isstrommen, Zachariae isbrae and Storsstrommen) (Joughin et al., 2010; Vallelonga et al., 2014). The drill camp was located at 75°38' N and 35°58' W in 2015, 2704 m a.s.l., 440 km to the South-East of the North Greenland Eemian Ice Drilling (NEEM) site. Ice flow velocity at the drill site is 55 m $a^{-1}$ (Hvidberg et al., 2020). Drilling has started in

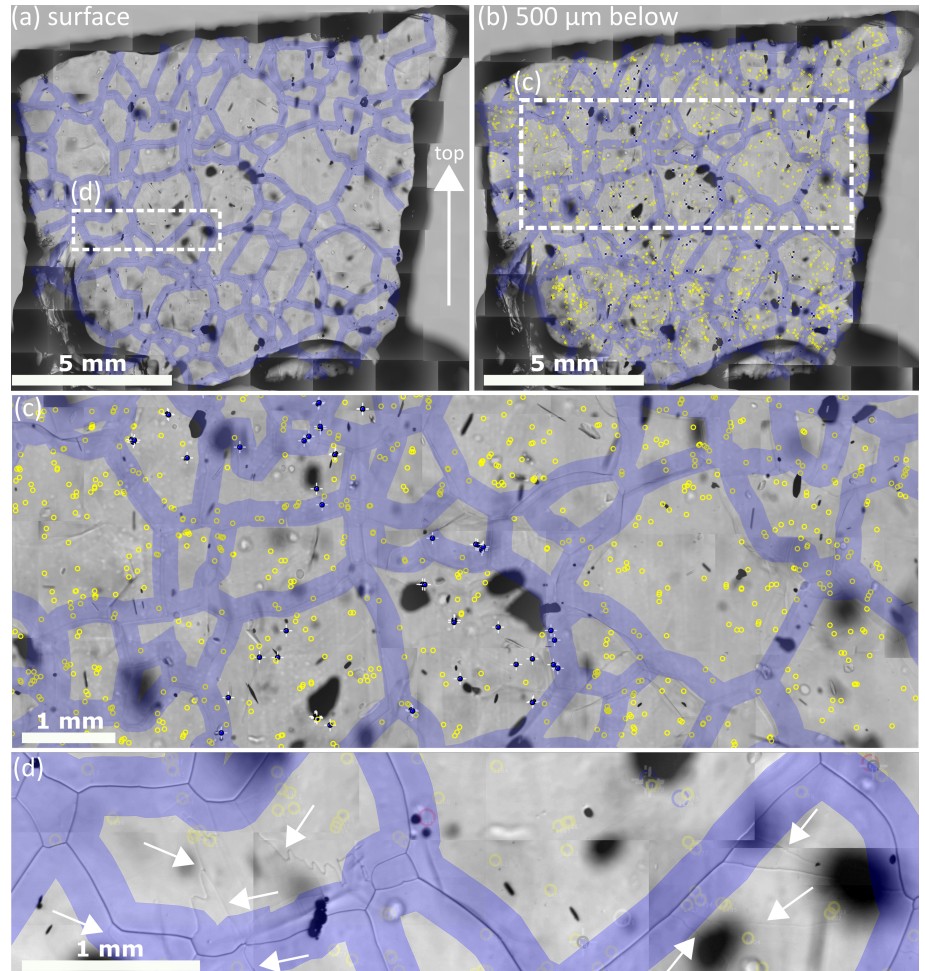

**Figure 2.** Details of the analysis procedure on the sample from a depth of 1339.75 m. Grain boundaries are indicated with 300 $\mu$m thick violet lines. Localised micro-inclusions 500 $\mu$m below the surface are indicated by yellow circles. Micro-inclusions analysed with Raman spectroscopy by Stoll et al. (2021a) are indicated by blue circles with white crosses. Out of focus black shapes are air bubbles. A) Map of the sample surface with highlighted grain boundaries, the arrow indicates the surface of the ice sheet. B) Impurity map with a focus depth of 500 $\mu$m below the sample surface, micro-inclusions and grain boundaries are indicated. C) Detail of the area indicated in B. D) Detail of the area indicated in A with different types of subgrain boundaries indicated by white arrows.

2016 and has been continued in the summer seasons of 2017, 2018, and 2019. The 2019 drilling season stopped at a depth of 2121 m, approximately 530 m above bedrock. At EGRIP the brittle ice zone is between 550 and 1000 m of depth according to Visual Stratigraphy and the core break record. Following Walker et al. (2018) the Holocene (present-11.7 ka) is in the upper 1240 m, the Younger Dryas (11.7-12.8 ka) at 1240-1280 m, and the Bølling Allerød (12.8-14.7 ka) at 1280-1375 m (Mojtabavi et al., 2020). We focus on the analysis of the upper 1340 m in this study.

## 2.2 Physical properties measurements

At EGRIP every 5-15 m fabric and microstructure measurements were performed continuously on 55 cm long sections ("bags"). Thick and thin sections (~90 mm length x 70 mm width x 0.3 mm thickness) were prepared and measured in situ in the EGRIP trench (-18°C). Thin sections were measured in 20 $\mu$m resolution with an automated fabric analyser (Wilson et al., 2003) (FA G50) from Russell-Head Instrument and grain size and CPO data (Fig. 1A) were obtained by digital image processing (Eichler, 2013). Grain sizes in each thin section were derived by the automated detection of grain boundaries and grains. The measured grain cross-sectional area is the number of pixels forming one grain (minimum grain size: 500 pixels) as described by Eichler (2013). The average grain size per section was used to determine the arithmetic mean grain size of each bag.

## 2.3 Sample choice and preparation

Samples and specific regions of interest for microstructural impurity analysis were defined using CFA (Stoll et al., 2021a), grain size and crystal orientation data (Fig. 1) with the aim to give an overview of the Holocene. Samples with high dust particle concentrations while simultaneously including different microstructural and fabric properties (e.g., small and large grains and different c-axes orientations) were chosen. We analysed ten samples in detail between depths of 138.92 and 1339.75 m (Fig. 1 and Table 2). The nine shallower samples are from the Holocene and the deepest sample is from the last glacial termination, i.e. the Bølling Allerød (Mojtabavi et al., 2020).

We used the remaining ice of the physical properties samples analysed at the EGRIP camp and followed the standard procedure by Kipfstuhl et al. (2006) to create thick sections with a thickness of ~10 mm. Contrary to Kipfstuhl et al. (2006) we did not use silicon oil, which produces intensive Raman spectra and can mask the micro-inclusion spectra. Different sample sizes were used, but most samples were approximately 10 x 10 mm (Table 2) and the surface and bottom of the samples were polished with a Leica microtome. Each polishing was followed by 1.5-2 hours of sublimation under controlled temperature and humidity conditions to obtain a good sample surface quality. Thus, grain-boundary grooves became more distinct and were easier to detect while small-scale disturbances (e.g., microtome scratches) were erased. Flawless surfaces enable the localisation of micro-inclusions inside the sample (500 $\mu$m below the surface) to produce high-quality microstructure maps.

## 2.4 Microstructure mapping and impurity maps

Microstructure mapping was performed following Kipfstuhl et al. (2006) and Eichler et al. (2017). Samples were placed under an optical Leica DMLM microscope with an attached CCD camera (Hamamatsu C5405), a software-controlled x-y stage and a frame grabber. The scanning resolution was 3 $\mu$mpix$^{-1}$ and several hundred individual photomicrographs were created grid-wise. This enabled the creation of high-resolution maps to detect micro-inclusions. These maps also provide the basis for a structured Raman spectroscopy study proving that the mapped dots are chemical impurities, i.e. micro-inclusions (Stoll et al., 2021a). Micro-inclusions are dust particles, droplets and salts inside the ice matrix and probably the most common form of impurity incorporations in ice. They are of the size of typical dust particles (1-2 $\mu$m) (Wegner et al., 2015) and are close to the resolution limit of the microscope. Transmission light mode and different focus depths enabled us to focus inside the ice

and to locate micro-inclusions below the surface (Fig. 2). The optical resolution of the photographs hampers the identification of the shape or volume of micro-inclusions. Locations of micro-inclusions were mapped manually following Eichler et al. (2017) which provides an "impurity map". Other features in the ice were visually identified as plate-like inclusions (caused by relaxation) (Nedelcu et al., 2009) and gas inclusions (air bubbles and clathrate hydrates) (Ohno et al., 2010; Weikusat et al., 2012) (Fig. 5C).

Impurity maps enable a structured and fast localisation of micro-inclusions with a confocal Raman microscope, which otherwise would be tedious in impurity-poor Holocene ice. They further allow the identification of micro-inclusions in the microstructure and thus preserve important spatial information. Grain boundaries were mapped on the sample surface and translated to the impurity map (Fig. 2). We applied a grain boundary width of 100, 200, and 300 $\mu$m and chose 300 $\mu$m as our reference for further analysis to compensate for light diffraction with depth and vertically tilted grain boundaries as done by Eichler et al. (2017). We created microstructure maps of all samples and located micro-inclusions in all of them. Micro-inclusions located in the 300 $\mu$m large grain boundary area are classified as "in the vicinity of the grain boundary" and serve as upper limit assumptions.

Small grains might enhance the probability of micro-inclusions being located close to grain boundaries. To compare samples with different grain sizes we measured the total area occupied by grain boundaries per sample for ten samples resulting in the ratio of micro-inclusions in the vicinity of grain boundaries to grain boundary area ($R_{GB}$):

$$R_{GB} = \frac{I_{GB}}{A_{GB}} \tag{1}$$

$I_{GB}$ is the percentage of micro-inclusions in the vicinity of grain boundaries, and $A_{GB}$ is the accumulated area occupied by grain boundaries per sample in percent using the upper limit assumption. A ratio of 1 implies a coherent relation of micro-inclusions in the vicinity of grain boundaries, while $R_{GB}<1$ implies less micro-inclusions in the vicinity of grain boundaries than assumed from the grain boundary area of the sample, $R_{GB}>1$ implies the opposite, i.e. more micro-inclusions in the vicinity of grain boundaries than implied by the grain boundary area. Furthermore, we performed a two-sided binomial test to derive the statistical significance of micro-inclusions being located in the vicinity of grain boundaries. The number of micro-inclusions in the vicinity of grain boundaries and the area occupied by grain boundaries were used to calculate the respective p-value.

## 3 Results

### 3.1 Evolution of grain size, CPO and microstructure with depth

We derive a profile of the grain size with depth, displaying the grain size evolution of the upper 1340 m of the EGRIP ice core and microstructural data from the depth regimes analysed with optical microscopy (Fig. 1).

**Grain size:** The mean grain sizes derived from the EGRIP core per 55 cm bag vary between 2.21 and 10.8 $mm^2$ (Fig. 1A). Starting from 3.9 $mm^2$ at the depth of 111 m, it steadily increases in the shallowest part and peaks around 500 m. Grain size

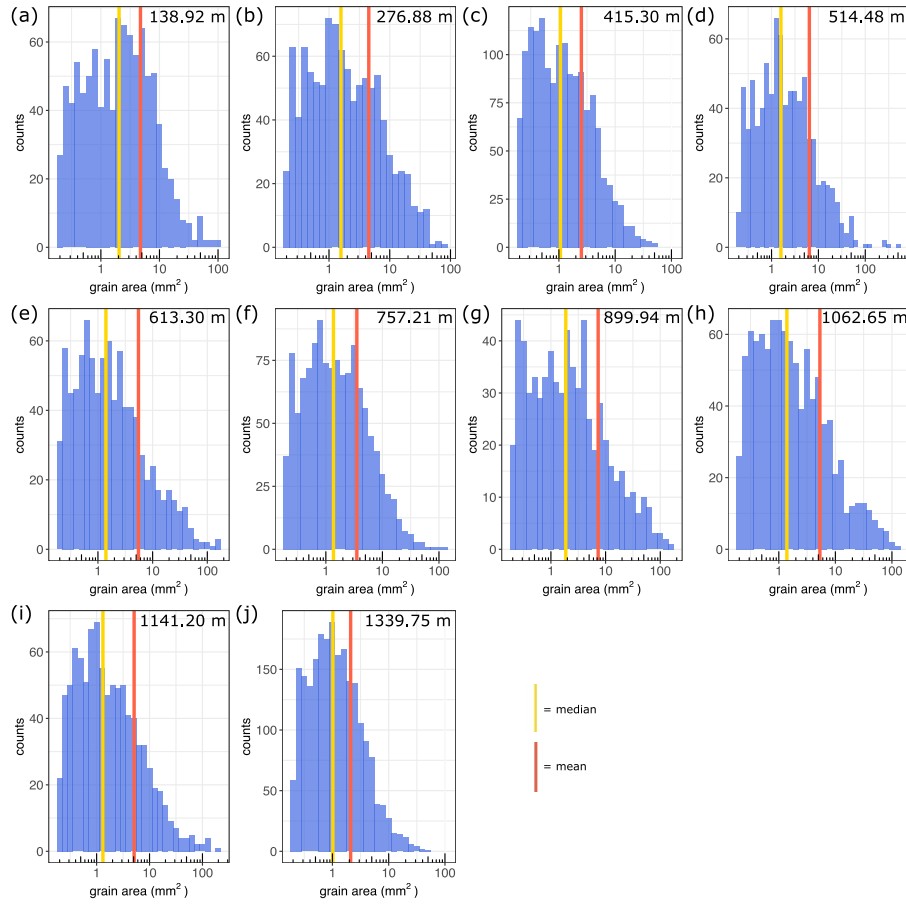

**Figure 3.** Grain size distribution within the 9 cm long physical properties samples. The indicated depths refer to the middle of the sample. Note the varying counts on the y-axis.

decreases uniformly until 900 m, where it remains at ~4.75 $mm^2$ until 1100 m of depth. The following 260 m are characterised by a steady decrease towards the minimum grain size of 2.21 $mm^2$.

We find the grain size to be highly variable on the centimetre-scale, mean values from neighbouring bags can vary up to 5 $mm^2$ (e.g., at 240 and 540 m). This occurs primarily in the shallower part of the Holocene ice, mean grain size values are more similar towards the Last Glacial (Fig. 1A). Fabric images (Fig. 1B-K) display the high variety of grain size on the centimetre-scale. Some samples show layers of very fine grains (e.g., Fig. 1F, G and K).

     The grain size distribution within the ten analysed thin sections is displayed in Fig. 3. Median grain sizes vary between 175   1.02 (1339.75 m) and 2.04 $mm^2$ (139.92 m) and mean grain sizes range from 2.11 (1339.75 m) to 7.31 $mm^2$ (899.98 m). All samples show an uneven, skewed right distribution with a tail towards larger grain sizes; mean values are higher than medians.

     **Crystal preferred orientation:** The analysed samples show a clear CPO evolution with depth. The shallowest sample shows a broad single maximum, which develops into a vague vertical girdle at 276 m. With depth, the girdle increases in strength and

is a fully developed vertical girdle below 1141 m (Fig. 1). Numbers of c-axes per sample vary between 673 and 1536 in the upper 1141 m. The deepest sample from a depth of 1339.75 m has 2232 measured c-axes orientations as a result of the small grain size in the last Glacial Termination and a thus higher number of grains per sample.

**Microstructure:** Common microstructural features in the upper 1340 m of the EGRIP ice core are subgrain boundaries (border regions of slightly different misorientations), bulging grains, grain islands (new grains inside highly distorted parent grains), and irregular shaped grains. Examples of these features from all depths are shown in Fig. 4. Subgrain boundaries are distributed heterogeneously in most grains in different intensities and shapes. Shapes range from normal or parallel to the basal plane to irregular zigzag patterns. These different types of subgrain boundaries often occur in the same grain. A few strongly developed subgrain boundaries were observed to cross grains completely independently of grain shape. Large grains usually inhibit more subgrain boundaries than small grains. Most grains are sharply or smoothly curved and grains often bulge towards grains with higher amounts of subgrain boundaries. Occasionally well developed (sub-)grain islands (Fig. 4G) were observed, mainly in large grains with protruding grain boundaries.

## 3.2 Localisation of micro-inclusions

In total, 5728 micro-inclusions (small yellow circles in Fig. 5) were located within ten analysed samples. In total, the spatial distribution of micro-inclusions in EGRIP ice is highly variable and micro-inclusions are heterogeneously distributed (Fig. 7). They have been found, solitary or in clusters, in the vicinity of grain boundaries, triple junctions and in the grain interior (Fig. 5). Distinct spatial distributional patterns have been observed throughout the core in varying strength and number, but no clear trend regarding preferred locations could be derived. No relationship between micro-inclusion distribution and depth was observed. However, the spatial distribution differs significantly depending on the examined scale as displayed in Fig. 5 and Fig. 7.

On the centimetre-scale layers of high micro-inclusion concentration were observed. However, regions with few micro-inclusions were also observed, e.g., at 415.3 m (see Table 2 column $n_{mi}$). On the millimetre-scale a variety of spatial distribution patterns was observed. Micro-inclusions form clusters (defined as three or more very close inclusions) and chains (Fig. 5B and C), which can extend over grain boundaries. Isolated inclusions were found in different distances to other micro-inclusions, ranging from micrometers to several millimetres (sometimes across grain boundaries). On the micrometre-scale several spatial patterns of micro-inclusions were observed. Inclusions were found in clusters of up to 4-5 single inclusions, in horizontal bands of varying lengths and at remote locations far away from other inclusions (Fig. 5B and C). Furthermore, large differences in the spatial distribution of micro-inclusions were observed from grain to grain. Grains with a high number of micro-inclusions are often neighboured by grains with only a few, or even zero, micro-inclusions despite being at the same depth. The number of micro-inclusions does not seem to be related to the orientations of the crystals. Micro-inclusions are seldom close to prominent microstructural features such as subgrain boundaries, grain islands or bulging grain boundaries. Micro-inclusions are rarely found on subgrain boundaries, which occur more often in larger grains.

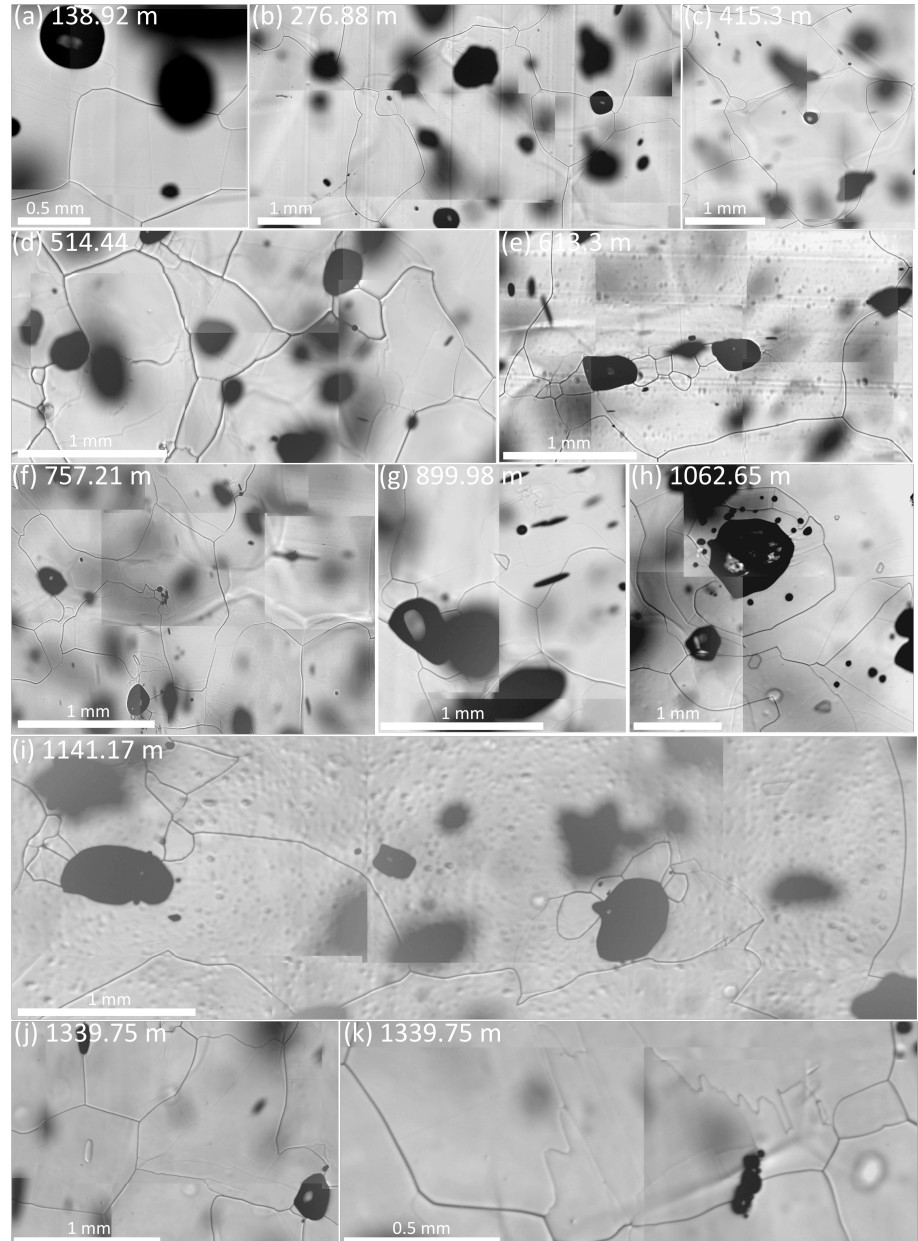

**Figure 4.** Microstructural observations from all depths. Dark and grey lines are grain and subgrain boundaries, respectively. Visible are bulged and cuspidate grain boundaries (A-K), different types of subgrain boundaries (A-K), and small grains at grain boundaries (C, D, E, I, J, K). Grain boundary pinning by bubbles and subgrain boundaries (B, E, I, J) and grain islands (G) also occur. Images vary in grey values due to different light conditions, grey values of lines vary due to different depths of etch grooves.

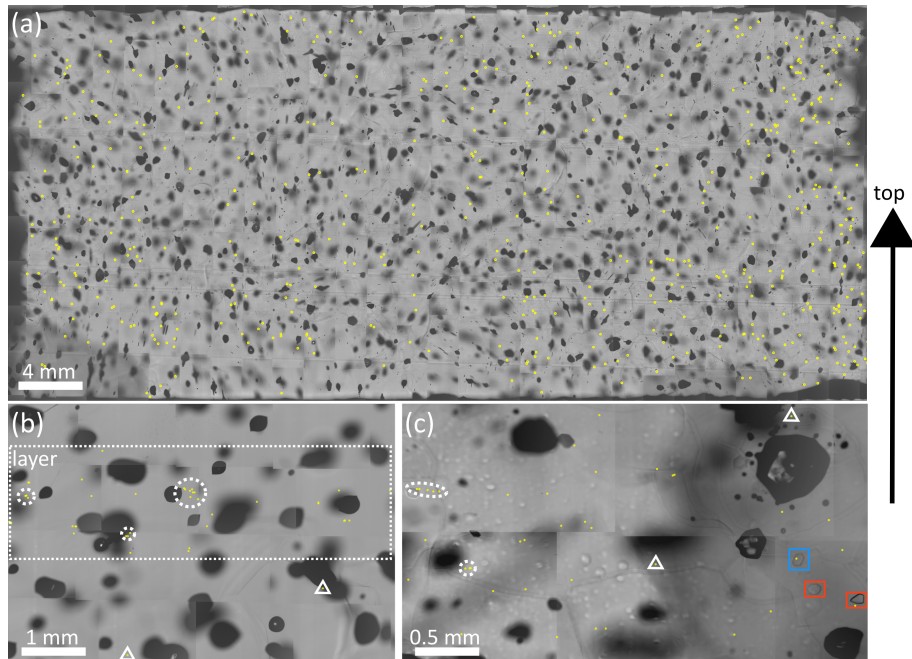

**Figure 5.** Locations of micro-inclusions 500 $\mu$m below the sample surface in three EGRIP samples. Annotations as in Fig. 2. White dotted circles in B) and C) indicate clusters or rows of micro-inclusions, white triangles indicate solitary micro-inclusions. Black shapes are bubbles. A) Overview of the micro-inclusions distribution at a depth of 613.3 m. Different localisation patterns can be observed, ranging from solitary inclusions to clusters of micro-inclusions (e.g., on the right side). B) Detailed section from a depth of 138.9 m. A layer of micro-inclusions, indicated by a white rectangle, accommodates clusters of micro-inclusions. C) Detailed section from a depth of 1062.65 m. Micro-inclusions are heterogeneously distributed. The blue rectangle indicates a plate-like inclusion and the red rectangles indicate clathrate hydrates.

### 3.3 Micro-inclusions in the vicinity of grain boundaries

We analysed the impact of the chosen grain boundary thickness by applying grain boundary thicknesses of 100, 200, and 300 $\mu$m to measure the number of micro-inclusions in the area occupied by grain boundaries (Table 1 and 2). For 100 $\mu$m the grain boundary area varies between 6.1% (138.92 m) and 13.4% (1339.75 m) while the number of micro-inclusions in this area varies between 5.8% (138.92 m) and 14.8% (1339.75 m) (Table 1). For 200 $\mu$m the grain boundary area varies between 11.6% (138.92 m) and 24.4% (1339.75 m) while the number of micro-inclusions in this area varies between 14.8% (514.44 m) and 31.4% (415.3 m). This results in $R_{GB}$ values between 0.7 (514.44 m) and 1.72 (415.3 m) and between 0.69 (613.3 m) and 1.78 (415.3 m) for 100 $\mu$m and 200 $\mu$m, respectively. The mean $R_{GB}$ of all samples increases slightly with grain boundary thickness from 1.09 (100 $\mu$m) to 1.15 (200 $\mu$m). From now on, we will always refer to a grain boundary thickness of 300 $\mu$m for consistency with Eichler et al. (2017). 1909 micro-inclusions were found in the proximity of 300 $\mu$m thick grain boundaries (Table 2 and Fig. 6). Our results are an upper limit assumption since real grain boundaries are basically interfaces,

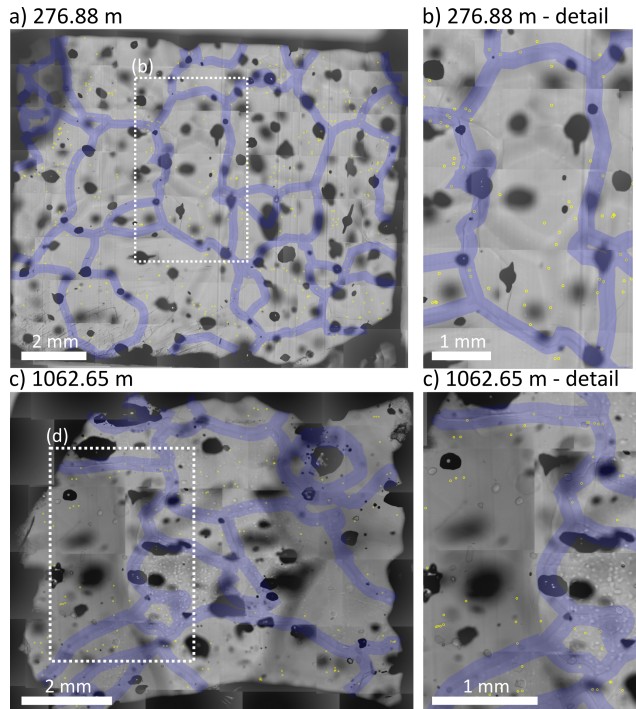

**Figure 6.** Samples with high $R_{GB}$ from different depths. Note the small-scale variety in the spatial distribution of micro-inclusions in regards to grain boundaries. Annotations as in Fig. 2. a) Overview of the sample at a depth of 276.88 m with $R_{GB}$=1.23. Inclusions tend to cluster. b) Detail of a) showing the difference between inclusions in the grain interior (centre) and in the vicinity of grain boundaries (upper left, lower right). c) Overview of the sample at a depth of 1062.65 m with $R_{GB}$=1.36. d) Detail of c) showing the difference between inclusions in the grain interior (left) and in the vicinity of grain boundaries (upper half).

with a discontinuity in lattice orientation accompanied by a zone of disorder of only a few molecule layers (Hobbs, 1974; Petrenko and Whitworth, 1999).

Between 22.3 and 42.4% of all micro-inclusions were found in the vicinity of grain boundaries; on average 33.3% of all micro-inclusions were in the vicinity of grain boundaries (Table 2, $I_{GB}$). The highest percentage of 42.4% was observed in the deepest sample from 1339.75 m of depth (Bølling Allerød). Grain boundaries cover between 18.7 and 43.2% of the sample area (Table 2, $A_{GB}$) resulting in an average grain boundary area of 25.9%. This parameter, an upper limit assumption and mainly controlled by the grain size, is close to the average value of our samples and less variable compared to the number of micro-inclusions in the vicinity of grain boundaries (Table 2, $A_{GB}$ and $I_{GB}$).

$R_{GB}$, calculated following Eq. (1), varies between 0.85 and 1.99, the mean ratio is 1.19. The lowest ratios of 0.85, 0.88 and 0.98 were found at depths of 514.44, 1141.17 and 1339.75 m, respectively. The highest ratios were 1.99, 1.40 and 1.36 at depths of 415.3, 757.21 and 1062.65 m, respectively. The deepest sample from a depth of 1339.75 m showed the highest number of micro-inclusions in the vicinity of grain boundaries (42.4%) and the largest area occupied by grain boundaries (43.2%) as here

**Table 1.** Micro-inclusions in the vicinity of grain boundaries with applied grain boundary widths of 100 and 200 $\mu$m. $A_{GB}$ and $I_{GB}$ were used in null hypothesis significance testing to calculate two-sided p-values.

| Depth (m) | $I_{GB_{100}}$ (%) | $I_{GB_{200}}$ (%) | $A_{GB_{100}}$ (%) | $A_{GB_{200}}$ (%) | $R_{GB_{100}}$ | $R_{GB_{200}}$ | p-value$_{GB_{100}}$ | p-value$_{GB_{200}}$ |
|---|---|---|---|---|---|---|---|---|
| 138.92 | 5.8 | 16.5 | 6.1 | 11.6 | 0.95 | 1.42 | 1.0 | 0.12 |
| 276.88 | 9.0 | 17.5 | 8.1 | 15.3 | 1.11 | 1.14 | 0.37 | 0.26 |
| 415.30 | 16 | 31.4 | 9.3 | 17.6 | 1.72 | 1.78 | 0.14 | 0.02 |
| 514.44 | 6.9 | 14.8 | 9.8 | 18.6 | 0.7 | 0.8 | 0.01 | 0.01 |
| 613.30 | 10.7 | 15.2 | 11.8 | 22.0 | 0.91 | 0.69 | 0.50 | 0.56 |
| 757.21 | 12.6 | 23.0 | 8.2 | 15.5 | 1.54 | 1.48 | $1.74 \cdot 10^{-5}$ | $1.78 \cdot 10^{-8}$ |
| 899.98 | 7.0 | 16.3 | 8.9 | 16.9 | 0.79 | 0.96 | 0.54 | 1.00 |
| 1062.65 | 13.5 | 25.6 | 10.4 | 19.3 | 1.3 | 1.33 | 0.25 | 0.08 |
| 1141.17 | 7.4 | 15.4 | 9.5 | 17.8 | 0.78 | 0.87 | 0.12 | 0.19 |
| 1339.75 | 14.8 | 30.9 | 13.4 | 24.4 | 1.1 | 1.26 | 0.14 | 0.06 |
| - | ∅ 10.37 | ∅ 20.66 | ∅ 9.55 | ∅ 17.9 | ∅ 1.09 | ∅ 1.15 | $\sum 3.11 \cdot 10^{-8}$ | $\sum < 2.2 \cdot 10^{-16}$ |

$I_{GB}$=percentage of micro-inclusions in the vicinity of grain boundaries, $A_{GB}$=accumulated area occupied by grain boundaries, $R_{GB}$=ratio of, $X_{GB100}$= grain boundary thickness of 100 $\mu$m, $X_{GB200}$= grain boundary thickness of 200 $\mu$m

**Table 2.** Samples and micro-inclusions statistics related to a grain boundary thickness of 300 $\mu$m in the upper 1340 m of the EGRIP ice core. $A_{GB}$ and $I_{GB}$ were used in null hypothesis significance testing to calculate two-sided p-values.

| Depth (m) | Age b2k (ka) | Sample size (mm x mm) | $n_{mi}$ | $n_{GB}$ | $I_{GB}$ (%) | $A_{GB}$ (%) | $R_{GB}$ | p-value |
|---|---|---|---|---|---|---|---|---|
| 138.92 | 1.0 | 13.01 x 16.96 | 103 | 23 | 22.3 | 18.7 | 1.19 | 0.375 |
| 276.88 | 2.2 | 11.00 x 12.76 | 342 | 85 | 24.9 | 20.2 | 1.23 | 0.037 |
| 415.30 | 3.5 | 10.01 x 13.27 | 51 | 21 | 41.2 | 20.7 | 1.99 | $8.2 \cdot 10^{-4}$ |
| 514.44 | 4.3 | 9.94 x 60.60 | 752 | 167 | 22.2 | 26.0 | 0.85 | 0.018 |
| 613.30 | 5.2 | 17.31 x 54.19 | 532 | 161 | 30.3 | 30.2 | 1.00 | 0.962 |
| 757.21 | 6.4 | 90.00 x 16.78 | 817 | 257 | 31.5 | 22.5 | 1.40 | $3.8 \cdot 10^{-9}$ |
| 899.98 | 7.6 | 11.32 x 12.62 | 129 | 30 | 23.3 | 23.0 | 1.01 | 0.917 |
| 1062.65 | 9.3 | 9.67 x 8.47 | 133 | 46 | 34.6 | 25.5 | 1.36 | 0.022 |
| 1141.17 | 10.2 | 16.61 x 21.67 | 486 | 109 | 22.4 | 25.5 | 0.88 | 0.131 |
| 1339.75 | 14.1 | 11.24 x 11.96 | 2838 | 1010 | 42.4 | 43.2 | 0.98 | 0.549 |
| - | - | - | $\sum 5728$ | $\sum 1909$ | ∅ 33.3 | ∅ 25.9 | ∅ 1.19 | $\sum < 2.2 \cdot 10^{-16}$ |

$n_{mi}$=number of localised micro-inclusions, $n_{GB}$=micro-inclusions in the vicinity of grain boundaries, $I_{GB}$=percentage of micro-inclusions in the vicinity of grain boundaries, $A_{GB}$=accumulated area occupied by grain boundaries, $R_{GB}$=ratio of grain boundary area to micro-inclusions in the vicinity of grain boundaries, b2k=before 2000 (Mojtabavi et al., 2020)

we have the smallest grain size of all inspected samples (Fig. 1A). In general, ratios do not vary much throughout our samples and are close to 1 indicating that the relative number of micro-inclusions in the vicinity of grain boundaries is comparable at all depths. The exceptionally high ratio of 1.99 at 415.3 m is most likely caused by the low total number of micro-inclusions (51) at this depth. Excluding this sample lowers the average $R_{GB}$ to 1.1.

$A_{GB}$ and $I_{GB}$ were used to perform null hypothesis significance testing for ten samples with an alpha of 0.05 as the cutoff for significance. The null hypothesis states that micro-inclusions are preferentially located in the vicinity of grain boundaries compared to the grain interior. If the p-value is smaller than 0.05, we reject this null hypothesis. Calculated p-values vary from sample to sample and range from 0.962 to $3.8 \cdot 10^{-9}$ (Table 2, p-value). Five samples have p-values below 0.05 while the other five samples have p-values above 0.05 (examples in Fig. 6). Thus, in 50% of our samples there are significantly more micro-inclusions in the vicinity of grain boundaries than expected, the majority are however located in the grain interior. A small p-value often correlates with a high $R_{GB}$ (except at 514.44 m) while a high p-value usually correlates with a low $R_{GB}$.

### 3.4 Identified micro-inclusions in the vicinity of grain boundaries

Here we present the mineralogy of micro-inclusions, identified with Raman spectroscopy (Stoll et al., 2021a), located in the vicinity of grain boundaries at all depths. In total, 181 of all identified micro-inclusions were located in the vicinity of grain boundaries (i.e. 22.9%). 92 sulphate particles were found in the vicinity of grain boundaries, followed by gypsum (47), quartz (28), mica (22), feldspar (21), nitrates (9), hematite (8), and titanite and anatase (both 1). 31.3% of all feldspar was found in the vicinity of grain boundaries, followed by 27.6% of all gypsum , 27.2% of all mica (Table 3), 23.8% of all sulphates (including gypsum, 20.8% without gypsum), and 22.2% of all quartz.

At depths with a high diversity in sulphate types (Fig. 4, Table 2, paragraph 3.3 in Stoll et al. (2021a)), i.e. at 138.92, 276.88, 514.44, 613.3, 757.21, and 899.98 m, the relative amount of sulphates in the vicinity of grain boundaries was higher than the relative amount of terrestrial dust, i.e. quartz, mica and feldspar. Below 900 m feldspar, mica and quartz were more common in the vicinity of grain boundaries than sulphates in the form of gypsum. However, at 1339.75 m 37.9% of all identified micro-inclusions in the vicinity of grain boundaries were gypsum.

## 4 Discussion

### 4.1 Evolution of grain size, microstructure and CPO at EGRIP

We briefly discuss the evolution of the mean grain size per 9 cm sample with depth, which will be analysed in more detail in a future study. The observed grain size evolution with depth is similar to findings from the NEEM ice core (Montagnat et al., 2014) (see below), but grain size can vary highly on the centimetre-scale. The high variability in grain size is also displayed in the grain size distribution (Fig. 3), which also reflects the visual identification of fine grains (Fig. 1B-K). The broad distribution of grain sizes is evidence for dynamic grain growth acting via dynamic recrystallisation, a process active during or prior/after deformation. Large grains are an indicator of ordinary boundary migration recrystallization (SIBM-O) while tiny grains indicate

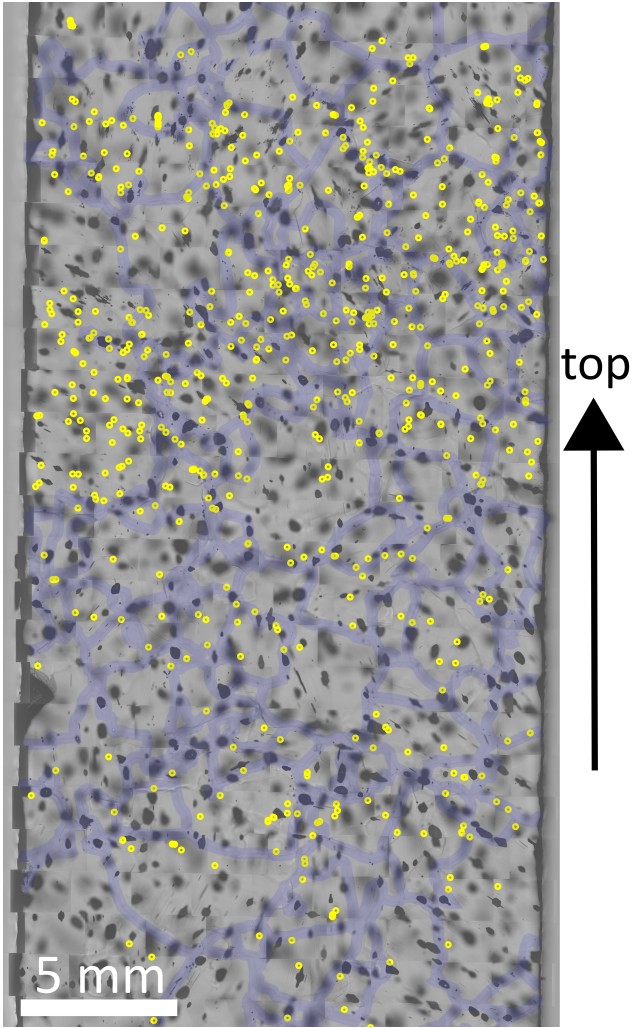

**Figure 7.** Locations of micro-inclusions at 757.21 m of depth. There is a cm-thick layer of inclusions in the upper half while the lower half is characterised by fewer inclusions and a heterogeneous distribution. Annotations as in Fig. 2.

nucleating migration recrystallization (SIBM-N) (nomenclature after Faria et al. (2014b)). Recrystallisation mechanisms are discussed in more detail in section 4.4.

A comparison with the NEEM ice core shows a similar grain size evolution with depth (Fig. 8) even though the dynamic settings are different (ice divide vs. ice stream). EGRIP grain size is generally larger in the upper 500 m and steadily decreases below this depth. Variability between samples from similar depths is small below 1200 m. NEEM grain size increases until a

depth of 740 m and is rather stable until 1350 m even though grain size variability is extreme such that is varies for several $mm^2$ between samples from similar depths. The cores are roughly 450 km apart, but show a similar depth-age relationship in the investigated depth range: the Glacial-Holocene transition is at 1375 m of depth at EGRIP (Mojtabavi et al., 2020) and

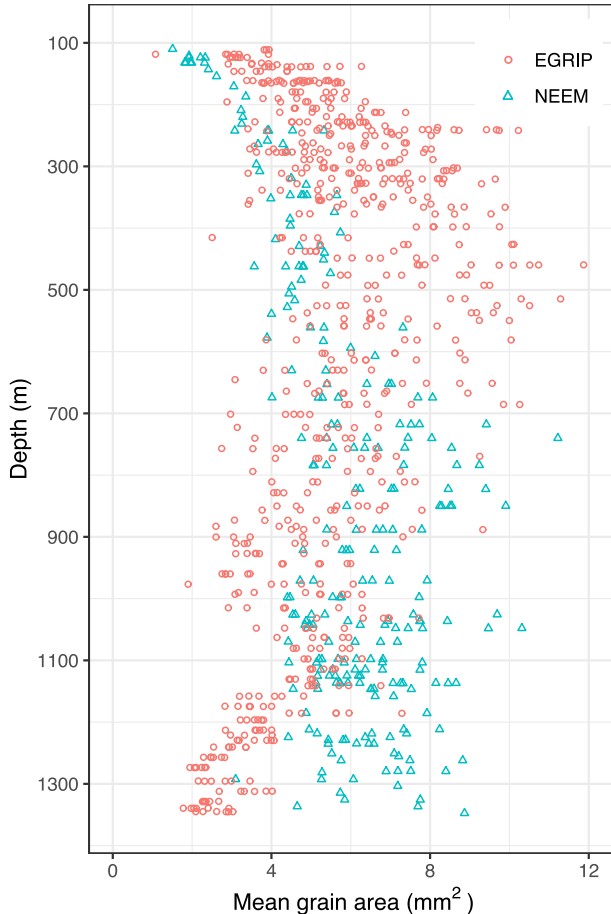

**Figure 8.** Grain size evolution at EGRIP and NEEM down to 1340 m. For EGRIP 9 cm section mean grain sizes derived via FA G50 are shown. NEEM data from Montagnat et al. (2014).

at 1420 m at NEEM (Rasmussen et al., 2013). The ice stream thus seems to have an impact on grain growth via dynamic recrystallisation in the upper several hundreds of meters. However, within the depth regime below down to the dust-loaded

Glacial ice the grain size evolution seems similar to NEEM and (so far) without observable effects by the ice stream.

The broad single maximum CPO indicates a random orientation of crystals in the shallowest part of the core comparable to other ice cores, such as NEEM (Montagnat et al., 2014), European Project for Ice Coring in Antarctica in Dronning Maud Land (EDML) (Weikusat et al., 2017a), and West Antarctic Ice Sheet (WAIS) (Fitzpatrick et al., 2014), and is probably the result of the just beginning vertical compression from overlying layers (Dahl-Jensen et al., 1997; Thorsteinsson et al., 1997;

Faria et al., 2014b) possibly preceded by temperature gradient metamorphism (Montagnat et al., 2020). With depth, crystals rotate and basal planes shift towards the direction of extension and produce vertical girdle CPOs (Thorsteinsson et al., 1997; Wang et al., 2002). This agrees with the observed surface flow pattern of NEGIS (Fahnestock et al., 2001; Joughin et al., 2017;

Hvidberg et al., 2020), visual stratigraphy and CPOs measured in EGRIP thin sections from the Last Glacial (Westhoff et al., 2020). Fabric images show a diversity in c-axis orientations and enhance the girdle formation.

Microstructural features are discussed in detail in Sec. 4.4. In-depth studies on the physical properties of the EGRIP ice core will follow and are a chance to enhance our understanding of ice stream dynamics.

## 4.2 Localisation of micro-inclusions

### 4.2.1 General findings

We localised more than 5700 micro-inclusions within ten samples from the upper 1340 m of the EGRIP ice core. Combining 290 optical microscopy and Raman spectroscopy confirmed that mapped micro-inclusions are indeed visible impurities below the sample surface supporting previous studies by Ohno et al. (2005, 2006); Eichler et al. (2017, 2019). The CFA data presented by Stoll et al. (2021a) support our micro-inclusion counts since CFA dust particle peaks correlate with areas of highly abundant micro-inclusions.

### 4.2.2 Localisation as found with microstructure mapping and Raman spectroscopy

Stoll et al. (2021b) found that the observed locations of impurities in polar ice are highly diverse and results seem to be strongly influenced by the applied method. Similar to other studies applying Raman spectroscopy (e.g., Ohno et al., 2005; Sakurai et al., 2009; Eichler et al., 2017, 2019) micro-inclusions are heterogeneously distributed throughout our samples and the number of micro-inclusions close to grain boundaries varies from sample to sample. However, only a small number of micro-inclusions is located directly on grain boundaries, the majority are located close to them. To quantitatively compare our 300 results we calculated the ratio of micro-inclusions in the vicinity of grain boundaries to the area of grain boundaries per sample, which reduces the impact of grain-size. Our estimates are likely too large due to the exaggerated thickness of grain boundaries, which are much smaller than 300 $\mu$m. The ratio $R_{GB}$ oscillates around 1, which describes, on average, a coherent distribution and a homogeneous spread of micro-inclusions in relation to grain boundaries throughout the core. $R_{GB}$ tends slightly towards larger values indicating that micro-inclusions are slightly more often located in the vicinity of grain boundaries than presumed 305 from the grain boundary area. The derived p-values (Table 2, p-value) support these findings, but also emphasise the high variability between samples. The localisation of solid micro-inclusions seems to be much weaker than recently observed for dissolved impurities (Bohleber et al., 2020, 2021).

### 4.2.3 Localisation and methodology

So far, there is only a small number of studies with a comparable quantitative approach regarding impurities in the vicinity 310 of grain boundaries (e.g., Eichler et al., 2017). However, it must be kept in mind that optical microscopy and Raman spectroscopy only allow the detection of visible micro-inclusions. We cannot make assumptions about the locations of dissolved, i.e. invisible, impurities. Early studies (e.g., Cullen and Baker, 2000; Barnes et al., 2002b; Baker and Cullen, 2003) and recent

**Table 3.** Minerals located most often in the vicinity of grain boundaries in relation to their absolute numbers and their crystal structure information in comparison to ice.

| Mineral | Inclusions at GB (%) | Axial lengths (Å) | Interaxial angles (°) | Crystal system |
|---------|---------------------|-------------------|----------------------|----------------|
| Feldspar | 31.3 | a=8.290, b=12.966, c=7.151 | $\alpha = 91.18$, $\beta = 116.31$, $\gamma = 90.16$ | triclinic |
| Gypsum | 27.6 | a=5.674, b=15.105, c=6.491 | $\alpha = 90$, $\beta = 118.51$, $\gamma = 90$ | monoclinic |
| Mica | 27.2 | a=5.386, b=9.324, c=10.268 | $\alpha = 90$, $\beta = 100.63$, $\gamma = 90$ | trigonal |
| Ice 1h | - | a=4.497, b=4.497, c=7.321 | $\alpha = 90$, $\beta = 90$, $\gamma = 120$ | hexagonal |

GB=grain boundaries, minerals at GB data from Stoll et al. (2021a), crystal structure data from Hudson Institute of Mineralogy (2021).

LA-ICP-MS results by Della Lunga et al. (2017); Bohleber et al. (2020, 2021) show that some soluble impurities, especially Na, seem to be preferably located at grain boundaries of Antarctic ice.

The comparison of different grain boundary thicknesses (100, 200, and 300 $\mu$m in Table 1 and Table 2) shows a similar ratio of micro-inclusion in the vicinity to grain boundaries and grain boundary area, i.e. $R_{GB}$, independent of the chosen grain boundary thickness or sample. This indicates that the relative number of micro-inclusions in the vicinity of grain boundaries is, on average, relatively stable and not significantly impacted by the chosen grain boundary thickness. We do not know the inclination of the grain boundaries below the surface, which supports the use of a 300 $\mu$m thick grain boundary which allows
an inclination of the grain boundaries of up to 33.3°.

The sample from 1339.75 m of depth is from the Bølling Allerød, a period with dust concentrations comparable to the Last Glacial Maximum, and thus explains the high numbers of dust particles and micro-inclusions. Visual stratigraphy at this depth (Weikusat et al., 2020) already shows cloudy bands, which are characterised by fine grain size and a thus large area occupied by grain boundaries (Gow and Williamson, 1971; Svensson et al., 2005; Faria et al., 2010). These bands are
probably caused by depositional events such as precipitation or sastrugi formation by redistributed surface snow (Svensson et al., 2005). Investigating those bands in detail from a microstructural and impurity perspective would help to better understand the relationship between impurity content, grain size and deformation.

Eichler et al. (2019) suggest that a higher number of micro-inclusions over a larger area should be analysed for improved statistics and more quantitative comparisons. We mostly succeeded in this approach by measuring a large total number of
micro-inclusions and, on average, more micro-inclusions per sample compared to similar studies (Ohno et al., 2005, 2006; Sakurai et al., 2010, 2011; Ohno et al., 2014). Measuring time is still an issue, especially when taking into account processes affecting the sample (e.g., sublimation). Other technical limitations are the resolution and contrast of the optics preventing the detection of very small micro-inclusions. However, the vast majority of the total micro-inclusions should have been counted since most detectable micro-inclusions have an average diameter of 1 to 3 $\mu$m (Ruth et al., 2003; Sakurai et al., 2009; Wegner
et al., 2015), and are thus within the resolution range of optical microscopes. The sample from a depth of 415.3 m has to be treated with caution. It has the highest percentage of micro-inclusions at grain-boundaries (21 of 51, i.e. 41.2%), which can be explained by the small absolute number of micro-inclusions and is thus, most likely a statistical outlier. The comparably small

area occupied by grain-boundaries (20.0%) further supports the possibility of an outlier and a biased statistics caused by the small total number of micro-inclusions in this sample.

#### 4.2.4 Localisation of micro-inclusions at different depths

We rarely observed distinct horizontal bands of micro-inclusions except at a depth of 757 m (Fig. 7). 7-11% of all micro-inclusions in EDML ice analysed by Eichler et al. (2017) were in the vicinity of grain boundaries, these samples are from the early Eeminan (MIS 5.5), i.e. "warm period ice", and thus comparable to our Holocene samples. Their NEEM samples showed a more spread-out distribution across the samples, and between 18 and 24% of micro-inclusions were in the vicinity of grain

boundaries. This supports our findings (22-41% in the vicinity of grain boundaries) in EGRIP Holocene ice even though our values are generally higher. Eichler et al. (2017) analysed NEEM samples from a depth of 739.9 and 740.2 m, which is close to our samples from 613.3 and 757.21 m of depth with values of 30.3 and 31.5 %, respectively. Our higher value can be explained by sample selection from depths with high dust content. Our results indicate that the spatial variability of micro-inclusions within Holocene ice is larger than previously thought emphasising the difficulty in generalising spatial patterns as suggested

by Stoll et al. (2021b).

Summarising, our observations regarding the location of micro-inclusions show that the spatial variability is very high and changes on the millimetre- to centimetre-scale. Thus, generalisations about the location of micro-inclusions should be made with care and have to be specified for the depth intervals considered. However, the majority of micro-inclusions are found in the grain interior even though grain boundaries at some depths locate comparably high numbers of micro-inclusions. Interestingly,

we were not able to link physical properties of the ice, such as grain size and CPO, to distinct distributional patterns of micro-inclusions. Investigating these relationships remains challenging due to e.g., 1) the different sizes of ice grains and samples and 2) the heterogeneity in grain size and micro-inclusion distribution on the mm to cm-scale. We thus suggest in-depth investigations of cloudy bands for future research.

### 4.3 Impact of micro-inclusions on ice properties

Impurities affect the physical and mechanical properties of ice, especially the deformation and the flow of ice (Paterson, 1991; Cullen and Baker, 2001), and most studies show that impurity-rich ice is easier to deform than impurity-poor ice (e.g., Fisher and Koerner, 1986; Paterson, 1991; Cuffey et al., 2000). The potential influence of impurities on other, partly related, ice properties, such as CPO and grain size, are complex and manifold (Stoll et al., 2021b). We discuss potential processes involving micro-inclusions and observed microstructural features and their implications for deformation via dominant mechanisms, and

dynamic recrystallization.

Zener Rpinning and drag of grain boundaries by impurities are suggested to be major mechanisms impacting grain size (e.g., Smith, 1948; Alley et al., 1986a; Fisher and Koerner, 1986; Alley and Woods, 1996; Paterson, 1991; Weiss et al., 2002; Durand et al., 2006). Even though roughly one third of all mapped micro-inclusions were located in proximity (300 $\mu$m upper limit assumption) to grain boundaries (Table 2), we did not observe any direct indications of micro-inclusions involved in Zener

pinning or drag. Direct indications would be e.g., a sharp edge at a second phase particle in an otherwise curved grain boundary

segment or a bulging grain boundary restricted by a second phase particle (e.g., Fig. 2 in Stoll et al. (2021b)) (Passchier and Trouw, 2005). This agrees with other studies (Ohno et al., 2005; Faria et al., 2010; Eichler et al., 2017, 2019) and indicates that the grain evolution is not affected strongly by the strict Zener pinning process of micro-inclusions as proposed by Alley et al. (1986b), but if affected via reduced grain growth at all, it may rather be affected by a reduced grain boundary mobility.

Fisher and Koerner (1986); Alley et al. (1986b); Li et al. (1998); Iliescu and Baker (2008) suggest that impurity concentrations must be above a threshold to result in counteracted grain boundary mobility and restricted grain growth by Zener pinning. However, quantitative thresholds are vaguely defined and partly ambiguous and thus difficult to discuss. It is possible that EGRIP Holocene dust concentrations are below this vague threshold to impact grain size development significantly via pinning, but unlikely because we also analysed samples comparable to Glacial ice, i.e. with high dust concentrations and small grains

(e.g., Fig. 1J and K). However, to confirm this assumption, a larger number of samples from the glacial period must be analysed.

High impurity layers showed no microstructural evidence for grain boundary sliding, such as linked-up grain boundaries or rectangular and lath-shaped grains (Fig. 6 in Goldsby and Kohlstedt (1997), Fig. 2B in Kuiper et al. (2020b)). Our results (Fig. 4) indicate that solid micro-inclusions do not have a major impact on the grain size evolution by e.g., enhancing the gradients of internal strain energies due to localised deformation along grain boundaries. However, it is possible that further methodological

progress is needed to directly identify the impact of solid inclusions on grain size and potential localised deformation via other deformation mechanisms e.g., Frank-Read sources and the multiplication and entanglement of dislocations, heterogeneous strain distribution within grains or dislocation pile-up on inclusions representing glide obstacles (Frank and Read, 1950; Ahmad et al., 1986; Weertman and Weertman, 1992).

The localisation of micro-inclusions in the vicinity of grain boundaries in our samples seems to be somewhat related to

their mineralogy (Table 3). In the case of solid micro-inclusions the interface properties between the ice and the particle play a crucial role and are thus probably influenced by the mineralogy and thus the surface properties of the inclusion at a grain boundary. We know from anti-freeze proteins (e.g., Bayer-Giraldi et al., 2018) that "ice-binding" properties result from similar surface structures (on the crystal lattice scale) of the second phase particle and the ice crystal. Less than one third of all feldspar, gypsum, and mica inclusions were located in the vicinity of grain boundaries, which is in good agreement with our

average derived percentage of micro-inclusions in the vicinity of grain boundaries. These minerals have partly comparable crystallographic lattice parameters to ice Ih (Table 3). Especially the properties of feldspar are similar, or multiples, to the properties of ice while the crystallographic properties of gypsum and mica are less similar. Fenter et al. (2000) show that, after removing the outermost K ions, remaining Si atoms of K-rich feldspars become attached to OH or O resulting in a surface prone to interact with water molecules via hydrogen bonding. In addition, electrostatic interactions between the feldspar surface

and the dipole moment of water might be enabled by the charged crystal lattice of feldspars (Yakobi-Hancock et al., 2013). Investigating the impact of these crystallographic properties in depth goes beyond the scope of this study, but might be of interest for future studies.

While Bohleber et al. (2020) found element-specific localisation trends for dissolved impurities we show that it is more complex for solid micro-inclusions. While the composition of the micro-inclusions may play a certain role, as explained above,

it seems to become clear that its state, i.e. solid, is more important especially when compared to studies on dissolved impurities

(e.g., Bohleber et al., 2020). Bohleber et al. (2020) show that $Na^+$ intensity peaks at grain boundaries, as proposed by e.g., Barnes and Wolff (2004), which supports the probable difference between dissolved and undissolved impurities (Stoll et al., 2021b). A comparison of different methods by analysing the same samples is thus of great interest to clarify the role of the 1) state of impurities and 2) of the applied method. With respect to the interface processes mentioned above, another aspect is the poorly understood structure of grain boundaries in ice. The density anomaly of water, in contrast to metals or minerals, leads to molecules in grain boundaries being packed more closely than in the lattice and amorphous water veins, which can be either liquid-like (e.g., Mader, 1992) or solid-like, thus exist at grain boundaries (Azuma et al., 2012). Depending on the conditions, mainly change in temperature, but possibly also pressure and salt content, these veins can become liquid-like and thus allow premelting and "slippery" grain boundaries as suspected by Kuiper et al. (2020a). Comparable to air bubbles during normal grain growth, dissolved impurities are mobile enough to interact and affix on migrating grain boundaries (Azuma et al., 2012), thus influencing their migration mobility. The concentration of dissolved impurities at grain boundaries could thus increase with time resulting eventually in a decreasing grain boundary mobility. Azuma et al. (2012) observed an increase in the number of air bubbles at grain boundaries with time, the possibility of a time-dependent concentration of dissolved impurities at grain boundaries should thus be investigated in detail with an appropriate method, such as LA-ICP-MS. That solid inclusions, in an almost constant temperature regime, do not tend to affix to grain boundaries, is indicated by our finding that the relative number of mineral particles found at grain boundaries does not change significantly with depth (Table 2).

Deforming ice might enable palaeorecord alteration and enhance impurities to change locations with time and thus, mix and react in ice as suggested by e.g., Masson-Delmotte et al. (2010); Faria et al. (2010); de Angelis et al. (2013); Baccolo et al. (2018). Migrating grain boundaries, driven by grain growth and dynamic recrystallization, could transport dissolved accumulated impurities, which eventually react, form and precipitate as salt micro-particles. Furthermore, impurities located in shear bands, which undergo high strain might lead to increased mechanical mixing rates (Eichler et al., 2019). Our study supports the possibility of such reactions occurring already in comparably shallow ice. The post depositional movement of sulphate ions suggested by Barnes et al. (2003); de Angelis et al. (2013) could explain the high frequency of clusters (Fig. 5). In deep Vostok ice de Angelis et al. (2005) observed the relocation of mineral particles in large clusters related to grain boundary migration during extreme grain growth conditions. Our samples are from a different temperature regime and not from bottom ice and processes, such as abnormal grain growth, thus do not take place, but the overall mechanisms could be similar and should be investigated further. de Angelis et al. (2013) suggest that in EPICA Dome C bottom ice acid-salt interactions, ion relocation and salt formation occur in situ in relation with ice recrystallization, but secondary salt formation is limited to acids relocated at grain boundaries at the surface of primary salts in inclusions. A large inclusion (~800 $\mu$m) analysed with high resolution synchrotron X-Ray micro-fluorescence was a complex structure containing a variety of minerals and tens of particle aggregates formed by gathering and mixing of several liquid phases and solid particles (de Angelis et al., 2013). Oversaturation in residual pockets containing carbonate and calcium ions resulted in the precipitation of calcium carbonate. The observation that micro-inclusions in our samples, especially sulphates, form clusters indicates that these processes have to be considered in EGRIP ice as well. Such detailed investigations are very valuable, but highly complex and costly and thus not suitable for extensive studies on micro-inclusions at several depths of one ice core as aimed for in this study.

## 4.4 Microstructural indications of recrystallization and deformation

Apart from direct impurity effects there are also indirect effects which could have an influence on the grain size development and the microstructure of our samples. Grain shape and grain size are both also a product of recrystallization processes, such as rotation recrystallization, strain-induced boundary migration with or without nucleation, and normal grain growth. Nucleating migration recrystallization (SIBM-N) involves the formation of new grain boundaries, if no new grains are formed ordinary boundary migration recrystallization (SIBM-O) takes place (Faria et al., 2014b). Indications of recrystallization processes, such as abundant subgrain boundaries, grain islands and bulging grain boundaries, were observed at all depths (Fig. 4) and indicate dynamic recrystallisation during deformation (in contrast to static recrystallization occurring without or after deformation) (Passchier and Trouw, 2005). Subgrain boundaries are stable structures of localised distortions of the lattice (Faria et al., 2014b). The vast occurrence of subgrain boundaries indicates internal stresses, heterogeneous strains, and high dislocation densities, which result in the arrangement of dislocations in arrays (polygonization after Poirier (1985)) and can be interpreted as early stages of rotation recrystallization (Drury and Urai, 1990) leading to the formation of new grain boundaries. All three subgrain boundary types (normal, parallel, zigzag) described by Weikusat et al. (2009) were observed at all depths (Fig. 4) indicating a high mechanical anisotropy. Electron Backscatter Diffraction or X-ray Laue diffraction would be needed to identify possible active slip systems of the dislocations in our samples (Piazolo et al., 2008; Weikusat et al., 2011, 2017b; Chauve et al., 2017). In analogy to previous results from polar ice (NEEM, EDML) the high abundance of subgrain boundaries parallel to the basal plane could be basal twist boundaries, but also have the potential that internal stresses are sufficient to also produce non-basal dislocations. However, rotation axes data are needed to verify this for EGRIP. Furthermore, the grain boundary shapes indicate strain-induced boundary migration driven by different dislocation densities in neighbouring grains and thus the bulging of grain boundaries into grains with higher strain energy (Weertman and Weertman, 1992; Humphreys and Hatherly, 2004). At the same depth, i.e. under similar boundary conditions, fewer fine grains undergo subgrain boundary formation than larger grains. Weikusat et al. (2009) observed the same for EDML ice and suggested that subgrain formation is not related to different shear behaviour, which is often attributed to a high impurity content (Paterson, 1991). Accepting recrystallization as the major grain size control a possible interpretation of varying grain size with varying impurity load can be found in the recrystallization diagram suggested by Faria et al. (2014b): recrystallization processes, in their sum, dictated by the given boundary conditions (pressure or strain rate and temperature) lead to a "steady state grain size" (Steinbach et al., 2017), in rock-forming minerals characterised as stress dependent (paleo-piezometer, e.g., Karato et al. (1980); Rutter (1995); Stipp and Tullis (2003)), and possibly reflected by observations (e.g., Jacka and Li, 1994; Treverrow et al., 2012). Impurity load is another boundary control and can move the "ground level" of the steady state grain size. Thus, under macroscopic conditions (pressure, temperature, strain rate) another (smaller) steady state grain size would be reached by the system. The process behind may be the changed mix of recrystallization processes, or the changed grain boundary mobility by impurity load (see above).

The observed microstructure (Fig. 4) indicates dynamic recrystallization and thus active deformation by dislocation activity throughout the upper 1340 m of the EGRIP ice core. Active deformation probably continues with depth due to similar

deformation mechanisms taking place below 1340 m as indicated by vertical girdle CPOs (Westhoff et al., 2020). Dynamic recrystallization does not act only at specific depth regimes, but throughout the ice column as was observed in other deep ice cores from less dynamic sites (e.g., Durand et al., 2008; Weikusat et al., 2009; Faria et al., 2014b). A dedicated study on the microstructure throughout the EGRIP ice core, analysing a larger number of samples, is however needed to investigate the internal mechanisms taking place in detail.

## 5    Conclusions

We here derive the first systematic analysis of the microstructural location of micro-inclusions in Holocene and Late Glacial ice from the EGRIP ice core, the first deep ice core from a fast flowing ice stream. The analysis of grain size, CPO and microstructural features along the core was accompanied by an investigation of the locations of micro-inclusions with respect to grain boundaries. The spatial distribution of micro-inclusions is highly diverse, and strongly depends on the applied scale.
Micro-inclusions are found in centimetre-thick layers, in local clusters and rows, and in solitariness. We quantify the relationship between micro-inclusions and grain boundaries and show that roughly two-thirds of all micro-inclusions are located in the grain interior. However, in half of all samples, grain boundaries are slightly more preferred locations than presumed from the grain boundary area. No significant relationship between depth and micro-inclusion distribution in relation to the microstructure was found. The combination of optical microscopy and Cryo-Raman spectroscopy shows that minerals, such as feldspar, seem to be more likely to be located in the vicinity of grain boundaries which could be caused by the interface properties due to their crystallographic structures, inclusion-ice interface properties or the structure of grain boundaries in ice.

Extensional deformation is the dominant deformation regime in our samples. However, no direct effects of micro-inclusions on the microstructure, such as typical local pinning microstructures, were observed. Observed clustering of micro-inclusions could indicate post depositional movement of micro-inclusions, which would imply possible effects on ice crystals already at shallow depths. All samples showed extensive implications of dynamic recrystallization, such as bulging grains, grain islands and different types of subgrain boundaries implying high internal strains and stresses, but no direct link to the location of the micro-inclusions. Grain boundary mobility is possibly reduced (overall not locally) by dissolved impurities, changing interface properties or the grain boundary structure impacted by impurities. Observing and defining the impact of impurities on the deformation of ice continues to be a challenge, which might need new methodological approaches. Our systematic overview of micro-inclusions throughout a large part of one ice core is a step forward towards more holistic impurity and microstructure studies, and thus a better understanding of the physical properties of polar ice.

*Data availability.*    High-resolution impurity maps and CPO data are available at PANGAEA. Grain size data on PANGAEA is being updated.

*Author contributions.* Conceptualisation by NS, IW, and MH. Microstructure mapping and Raman methodology developed by NS, JE, and IW. Investigation and data curation by NS. Formal analysis by NS and TE. Funding acquisition for NS by IW. The original draft was written
by NS with assistance from all co-authors.

*Competing interests.* The authors declare that the research was conducted in the absence of any commercial or financial relationships that could be construed as a potential conflict of interest.

*Acknowledgements.* This work was carried out as part of the Helmholtz Junior Research group "The effect of deformation mechanisms for ice sheet dynamics" (VH-NG-802). We especially thank the EGRIP physical properties team, e.g., Johanna Kerch, Ina Kleitz, Daniela
Jansen, Sebastian Hellmann, Wataru Shigeyama, Julien Westhoff, Ernst-Jan Kuiper, Tomoyuki Homma, Steven Franke and David Wallis. We also thank all other EGRIP participants for logistical support, ice processing and fruitful discussions. EGRIP is directed and organised by the Centre for Ice and Climate at the Niels Bohr Institute, University of Copenhagen. It is supported by funding agencies and institutions in Denmark (A. P. Møller Foundation, University of Copenhagen), USA (US National Science Foundation, Office of Polar Programs), Germany (Alfred Wegener Institute, Helmholtz Centre for Polar and Marine Research), Japan (National Institute of Polar Research and
Arctic Challenge for Sustainability), Norway (University of Bergen and Trond Mohn Foundation), Switzerland (Swiss National Science Foundation), France (French Polar Institute Paul-Emile Victor, Institute for Geosciences and Environmental research), Canada (University of Manitoba) and China (Chinese Academy of Sciences and Beijing Normal University). TE and CMJ gratefully acknowledge the long-term financial support of ice core research at the University of Bern by the Swiss National Science Foundation (grant no. 200020_172506 (iCEP) and 20FI21_164190 (EGRIP).

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
