# Peer review of "Microstructure, Micro-inclusions and Mineralogy along the EGRIP ice core - Part 1: Localisation of inclusions and deformation patterns"

_The Cryosphere, 2021_

## Author Comment (AC1)

**Replies to referees Referee 1**

**Ref1:** The authors present a very methodical, comprehensive examination of solid micro- inclusions in polar ice at a range of depths that shows the potential for such a systematic approach to answer many outstanding questions about the role of impurities in ice structure and evolution. The work represents the first investigation of solid micro- inclusions in fast moving polar ice, and uses the methods outlined by Eichler et al. (2017) to construct impurity maps of over 5000 micro-inclusions and allows a robust statistical analysis of the frequency of location of micro-inclusions within the ice microstructure at a level that has not been obtained to date, and helps shed light on several long-speculated processes.

Future implications are particularly tantalizing...e.g., the examination of the impacts of mineralogy, grain boundary sliding, and precipitation and recrystallization on ice properties from the microscale to the mesoscale, which has been suggested in past literature, but with no prior methodology for proving definitively, is a very interesting consequence of this work.

The methods, results and conclusions are well articulated and presented, I have only a few minor specific comments and a small amount of very minor, technical corrections.

**Reply:** We thank the referee for various comments and ideas to improve the manuscript. We adopted almost all of the suggestions and e.g., present a grain size comparison with the NEEM ice core (see details below). The individual comments are addressed below.

**Specific comments:**

**Ref1:** Figures: I found the figures particularly compelling and interesting, especially Figures 1 and 2 (but all of the figures showing the variability of the micro-inclusions both within and outside of grain boundaries is very interesting).

Reply: Thank you very much.

**Ref1: Abstract, line 10:**

"Analysing the area occupied by grain boundaries in the respective samples shows that micro-inclusions are slightly more often located at or close to grain boundaries in half of all samples. Throughout all samples we find strong indications of dynamic recrystallisation, such as grain islands, bulging grains and different types of subgrain boundaries."

I think understand this sentence, but it is slightly confusing to read it. (Took me a couple of times for it to make sense). I think just rewriting it as, "In half of all samples, micro- inclusions are more often located at or close to the grain boundaries by a slight margin (in the areas occupied by grain boundaries). Not sure that last bit is needed.

Reply: This sentence is indeed confusing. We happily adopted your suggestion and changed it.

**Ref1:** Pg 5, Figure 2 caption (and throughout): on the last line here, and in other areas throughout the text, you state that something is "rarely close" to the micro-inclusions. Is it possible to define what you mean by "close" and is it just the 300 micron buffer surrounding the grain boundaries that defines what "close" is? **Reply:** The last line of the figure caption of Figure 2 only refers to the distance of micro-inclusions to subgrain boundaries. There is no universal definition of a Subgrain boundary in glaciology and we thus did not map them. The statement in the figure caption refers to careful visual inspections of the Subgrain boundaries in the derived impurity maps. In this case "close" means inclusions on the Subgrain boundary.

To clarify this we changed the caption to: "Micro-inclusions are rarely found on subgrain boundaries, which occur more often in larger grains." Mapping and analysing Subgrain boundaries in detail might be part of a future study.

For grain boundaries and inclusions "close" refers to the 300 micron buffer around the grain boundaries which we clarified throughout the text by changing "at grain boundaries" to "in the vicinity of grain boundaries".

**Ref1:** Pg 6, Line 114 what were the criteria for choosing samples, ie., what CFA values, grain sizes and orientation? For example, it seems like the Bolling Allerod period was targeted for sampling, and samples in the Younger Dryas and before and after the Holocene, but what other criteria were used to choose depths of

**interest?**

**Reply:** The sample choice was done in two major steps with the overall aim to reach a systematic (high-resolution) overview of one ice core (limited by the availability due to COVID-19).

1) Samples were chosen to give a good representation of the entire core down to the Glacial (depth-age relationship available until 1360 m only), i.e. roughly every 100 m. We were limited to available samples (many samples are still stored in the EGRIP camp and could not been retrieved due to COVID-19).

2) We focused on depths with high CFA insoluble particle concentration, which were also already analysed regarding their microstructure and fabric data. We chose areas of interest with comparably high insoluble particle content (if possible with a gradient in concentration) to increase the chance of being able to do reliable statistics. We furthermore wanted samples to represent different grain sizes (very small to large) and crystal orientations (random to girdle) throughout the core and (if possible) within the sample (e.g., medium sized grains with a layer of very fine grains). Showing and explaining the exact criteria and data for each sample would be too lengthy and we thus decided to show an overview plot (Fig. 1).

We extended the methods section in the following way: "Samples and specific regions of interest for microstructural impurity analysis were defined using CFA (Stoll et al., 2021a), grainsize and crystal orientation data (Fig. 1) with the aim to give an overview of the Holocene. Samples with high dust particle concentrations while simultaneously including different microstructural and fabric properties (e.g., small and large grains and different c-axes orientations) were chosen. We analysed ten samples in detail between depths of 138.92 and 1339.75 m (Fig. 1and Table 1). The nine shallower samples are from the Holocene and the deepest sample is from the last glacial termination, i.e. the Bølling Allerød (Mojtabavi et al., 2020)".

**Ref1:** Pg 8, Line 194, Is it worth discussing how it is determined if micro inclusions are plates or clathrate hydrates here? It is mentioned in the figure caption for Figure 4, but seems like there is some more details that could be added about that in the text.

**Reply:** We discussed this issue and decided not to go into more detail in the original manuscript, because of the different sizes of micro-inclusions and bubbles/hydrates. These additional features can clearly play a role regarding the deformation, but assumptions are difficult to make and would need a dedicated study/data set. However, we added a brief section to the main text mentioning and explaining these observations. An in-depth discussion goes beyond the scope of this manuscript, but is of interest for future studies (especially for deeper samples).

"Other features in the ice were visually identified as plate-like inclusions (caused by relaxation) (Nedelcu et al., 2009) and gas inclusions (air bubbles and clathrate hydrates) (Ohno et al., 2010; Weikusat et al., 2012) (Fig. 5C)."

**Ref1:** Pg 13, Line 240, Not sure if this is planned for the future work, or what exactly it would look like, but is it possible to plot grain size evolution of NEEM vs. EGRIP? That would be interesting to see. I understand there are likely limits that can be made in the intercomparison due to depth/age mismatches and differences in sample sizes and resolution, but the location of EGRIP over the ice stream vs. NEEM would be very interesting to see.

**Reply:** This is indeed an interesting comparison, we did it as far as possible and added it (see figure below). For a better comparison we also now use the 9 cm section means instead of the 55 cm means and thus change the first figure (see below). The general grain size evolution of both cores is similar down to the so far available depth. We added the comparison in the discussion in the following way:

"A comparison with the NEEM ice core shows a similar grain size evolution with depth (Fig. 7) even though the dynamic settings are different (ice divide vs. ice stream). EGRIP grain size is generally larger in the upper 550 m and steadily decreases below this depth. NEEM grain size increases until a depth of 740 m and is rather stable until 1350 m even though grain size variability is extreme such that is varies for several mm2 between samples from similar depths. The cores are roughly 450 km apart, but show a similar depth-age relationship in the investigated depth range: the Glacial-Holocene transition is at 1375 m of depth at EGRIP (Mojtabavi et al., 2020) and at 1420 m at NEEM (Rasmussen et al., 2013). The ice stream thus seems to have an impact on grain growth via dynamic recrystallisation in the upper several hundreds of meters. However, within the depth regime below down to the dust-loaded Glacial ice the grain size evolution seems similar to NEEM and (so far) without observable effects by the ice stream.

---

## Author Comment (AC2)

**Replies to referees**
**Referee 2**

**Ref2:** This paper presents some interesting observations of impurity locations on 10 samples extracted from the EGRIP core between 138 and 1339 m depth. The experimental protocol appears robust.

The main results coming from these observations are first that insoluble impurities (the ones observed here) are heterogeneously spread on the samples, and within the grains and second that the observed spatial distribution depends on the observation scale. A specific area, 300 microns distance from the detected GB is defined in order to test a specific location of impurities closer to GBs, but there is no clear signal of it on the observed samples. A "companion paper" is mentioned all along the paper and many of the results provided in this companion paper are evoked in the paper, and a large part of the discussion is based on these results. Furthermore, very little of the discussion stands on the results presented here. Most of it refers to other studies or studies that would be required to be able to improve the "story".
Based on these main considerations, I would suggest the authors to gather the presented observations to the companion paper in order to make a paper with noticeable results to comment and discuss. I am pretty sure that such a complete study would have a valuable impact in the field.
The present paper seem "too weak" on this aspect to provide a publication per-se.
**Reply:** We thank the referee for a thorough review and discussed all mentioned aspects in detail. Many valuable thoughts were raised, which help to improve the quality of the manuscript.  We agree in several parts and included most of the suggestions. For example, we added detailed data about the grain size distribution of the 10 physical properties samples the inclusion studies are based on and an expanded analysis on the impact of grain boundary thickness (varying it between 100, 200, and 300 μm).  This also results in a strengthened discussion which will be described in more details below.

**Ref2:** I also have a strong concern about the way the specific location of grain boundaries is treated in the paper.
It is first clearly stated that the authors consider an arbitrary selected area, of 300 microns thick, around GBs as being representative of a closeness to GB. Then, in the text, and it is particularly problematic in the discussion or in the conclusion, this area is referred to as "AT grain boundary". I think this is dangerous as it could be easily mis-interpreted and it should be replaced by "close to GB" or "in the area of GB" all along the text. Indeed, readers that will not go through the details of the study will likely omit the strong assumption that observations are not made ON or AT the exact GB, but in a nearby area. I would also suggest the authors to add a critical analysis of the impact of the chosen thickness value on the statistical results they provide.
**Reply:** We agree that grain boundaries are indeed a sensible topic. In the methods we explain the definition of "at the grain boundaries", i.e. the 300 μm thick area. To avoid confusion or wrong interpretations of readers skipping through the manuscript we changed "at grain boundaries" to "in the vicinity of grain boundaries" throughout the text.
We further extended the grain boundary analysis by investigating grain boundary area and micro-inclusions in these areas for grain boundary thicknesses of 100 and 200 μm (Table 3). We extended the methods section to: "We applied a grain boundary width of 100, 200, and 300 μm and chose 300 μm as our reference for further analysis to compensate for light diffraction with depth and vertically tilted grain boundaries as done by Eichler et al. (2017)."
This analysis shows a rather stable relationship between grain boundary area and the amount of inclusions in this area, i.e. in the vicinity of grain boundaries. We state throughout the text that the results for 300 μm are an upper limit assumption, it is not possible to give a quantitative estimation. In the discussion we add the following text: "The comparison of different grain boundary thicknesses (100, 200, and 300 μm in Table 3) shows a similar ratio of micro-inclusion in the vicinity to grain boundaries and grain boundary area, i.e. $R_{GB}$, independent of the chosen grain boundary thickness or sample. This indicates that the relative number of micro-inclusions in the vicinity of grain boundaries is, on average, relatively stable and not significantly impacted by the chosen grain boundary thickness. We do not know the inclination of the grain boundaries below the surface, which supports the use of a 300 μm thick grain boundary as an upper limit assumption which allows an inclination of the grain boundaries of up to 33.3°.

**Table 1.** Micro-inclusions in the vicinity of grain boundaries with applied grain boundary widths of 100 and 200 $\mu$m. $A_{GB}$ and $I_{GB}$ were used in null hypothesis significance testing to calculate two-sided p-values.

| Depth (m) | $I_{GB_{100}}$ (%) | $I_{GB_{200}}$ (%) | $A_{GB_{100}}$ (%) | $A_{GB_{200}}$ (%) | $R_{GB_{100}}$ | $R_{GB_{200}}$ | p-value$_{GB_{100}}$ | p-value$_{GB_{200}}$ |
|---|---|---|---|---|---|---|---|---|
| 138.92 | 5.8 | 16.5 | 6.1 | 11.6 | 0.95 | 1.42 | 1.0 | 0.12 |
| 276.88 | 9.0 | 17.5 | 8.1 | 15.3 | 1.11 | 1.14 | 0.37 | 0.26 |
| 415.30 | 16 | 31.4 | 9.3 | 17.6 | 1.72 | 1.78 | 0.14 | 0.02 |
| 514.44 | 6.9 | 14.8 | 9.8 | 18.6 | 0.7 | 0.8 | 0.01 | 0.01 |
| 613.30 | 10.7 | 15.2 | 11.8 | 22.0 | 0.91 | 0.69 | 0.50 | 0.56 |
| 757.21 | 12.6 | 23.0 | 8.2 | 15.5 | 1.54 | 1.48 | $1.74 \cdot 10^{-5}$ | $1.78 \cdot 10^{-8}$ |
| 899.98 | 7.0 | 16.3 | 8.9 | 16.9 | 0.79 | 0.96 | 0.54 | 1.00 |
| 1062.65 | 13.5 | 25.6 | 10.4 | 19.3 | 1.3 | 1.33 | 0.25 | 0.08 |
| 1141.17 | 7.4 | 15.4 | 9.5 | 17.8 | 0.78 | 0.87 | 0.12 | 0.19 |
| 1339.75 | 14.8 | 30.9 | 13.4 | 24.4 | 1.1 | 1.26 | 0.14 | 0.06 |
| - | $\emptyset$ 10.37 | $\emptyset$ 20.66 | $\emptyset$ 9.55 | $\emptyset$ 17.9 | $\emptyset$ 1.09 | $\emptyset$ 1.15 | $\sum 3.11 \cdot 10^{-8}$ | $\sum < 2.2 \cdot 10^{-16}$ |

$I_{GB}$=percentage of micro-inclusions in the vicinity of grain boundaries, $A_{GB}$=accumulated area occupied by grain boundaries, $R_{GB}$=ratio of, $X_{GB100}$= grain boundary thickness of 100 $\mu$m, $X_{GB200}$= grain boundary thickness of 200 $\mu$m

**Ref2:** P1, l18: I think it should be mentioned some of the pioneer works of paleoclimate reconstruction from ice cores, such as Lorius et al. 1985 or Petit et al. 1999 (the authors may know other pioneer work from other teams)
**Reply:** We agree and added both references and Watanabe et al. (2003).

**Ref2:** p2, l. 33-34 : About the impact of grain size on the deformation of ice. I think the statement is too simplistic. Situation is different in the case of ice deformed in the lab, for which experimental investigations were properly made and show that grain size only matters for transient creep (Duval and Le Gac 1980) while the minimum creep rate of secondary creep (therefore the Glen's law) or the compressive strength does not depend on grain size (Duval and Le Gac 1980, Jones and Chew 1981, 1983, Jacka 1984, 1994). An exception concerns the work by Goldsby and Kohlstedt (1997) that's dealing with very small grain sizes not relevant for natural ice on Earth. Concerning ice deformation in the conditions of the central part of polar ice sheets, most study related with the impact of grain size are modeling studies, or interpolations, and the grain size effect mostly comes from the fact that assumptions are made of grain-size dependent rheologies to match the model or interpolation results. Therefore this dependance is a pre-given assumption rather than demonstrated by results.
**Reply:** We totally agree that this statement is simplified and that the impact of grain size is complex and context-dependent, however more recent work (e.g., following the grain size-stress relation observed by Treverrow et al. 2012) show that this dependence is more than a pre-given assumption. It would go way beyond this manuscript to discuss this issue in detail and so far there is still a lack of distinct data. Thus the next sentence clearly states that "The relationship between CPO, grain size, and impurity content and how they impact deformation is still under debate (Stoll et al., 2021)."
To respect the facts and studies you mention, we added "… however, under certain boundary conditions, it is further impacted by…".

**Ref2:** Fig 1: I am not sure, but it seems to have an incoherency between the pole figure representation of the texture and the color-code of the microstructure shown. For instance, on Fig 1 (k), green color represents an orientation that is perpendicular to the girdle represented in the pole figure. If I am right it might just be a question of representations, but it would make sense to have them coherent with each others.
**Reply:** This is true and origins from the different ways the ice core was rotated during ice core processing and cutting. Since they have all been rotated differently the orientations look different from the colour code, but the CPO is the same for all samples below ~400 m: a vertical girdle.

**Ref2:** Fig 2: Why is the last sentence here in the legend? It is a result analysis that should, to my point of view, remain in the text.
**Reply:** We agree and put in the results.

**Ref2:** Part 2.2: please provide the spatial resolution used for your AITA measurements.
**Reply:** That´s an important point we missed to include, a resolution of 20 μm was used.
We changed the text to "Thin sections were measured in 20 μm resolution with an automated fabric analyser…"

**Ref2:** Part 2.3: why did you study only one sample of the last glacial period? It seems too weak for a statistical comparison with the other period, and in the mean time, it is complicated to compare this single sample with the 9 others owing to there different climatic origin.
**Reply:** Due to the ongoing COVID-19 pandemic it has not been possible to drill all the way to the bedrock, a large part of the glacial is thus still missing. Furthermore, several samples are still stored at the EGRIP site and are thus not available for analysis. We thus decided to do a high-resolution study of the Holocene, which has not been done before in this form. The deepest sample is from the Bolling Alleród and thus from the transition to the main part of the glacial. Another reason is the available age record, which is so far only provided for the upper 1383 m. It is not enough for a full comparison of Holocene vs. Glacial ice (which we do not claim), but it is nevertheless interesting to show and discuss.

**Ref2:** P7 l. 142-143: here starts the problem with classifying "AT the grain boundary" the inclusions that are, in fact, in an area nearby a GB, whose size was arbitrary chosen to be 300 microns. Please replace by "nearby" or "in the vicinity of", etc. By the way, it would be good to justify this value of 300 microns?
**Reply:** To clarify this issue and to avoid the wrong impression when only jumping through the manuscript we exchanged "at grain boundaries" with "in the vicinity of" throughout the text when taking about our data.
The value is chosen to allow for the unknown inclination of the grain boundaries below the sample surface. With a focus 500 microns below the sample surface this allows an "angle of tolerance" of 33.3°. We also chose this value to allow a comparison with the results from Eichler et al. (2017). For samples with much smaller grains, e.g. from the Glacial, this would have to be adjusted as explained in our comment introducing the values for grain boundary thicknesses of 100 and 200 microns.

**Ref2:** P7 l. 146: I would call R_GB "the ratio of micro-inclusions to grain boundary area" and not the contrary?
**Reply:** We agree and changed the definition.

**Ref2:** P7 About grain size: considering the fact that you mention the grain size distributions to be wide, and very likely far from a gaussian, I think it would be more correct and more informative to provide a median, and a distribution (at least for a few samples). I know that mean area has been classically used, so it would be OK for me to keep giving this info, but I thinks it's time for us to be more correct in the way we present grain size values.
**Reply:** We fully agree and added an additional figure showing the grain size distribution of the physical properties samples (see below). We here include the mean and median values of each of the 10 samples. Hence, we added some text to the results:" The grain size distribution within the ten analysed thin sections is displayed in Fig. 3. Median grain sizes vary between 1.02 (1339.75 m) and 2.04 mm$^2$ (139.92 m) and mean grain sizes range from 2.11 (1339.75 m) to 7.31 mm$^2$ (899.98 m). All samples show an uneven, skewed right distribution with a tail towards larger grain sizes; mean values are higher than medians."

We discuss this as follows: "The comparison of different grain boundary thicknesses (100, 200, and 300 μm in Table 1 and Table 2) shows a similar ratio of micro-inclusion in the vicinity to grain boundaries and grain boundary area, i.e. R$_{GB}$, independent of the chosen grain boundary thickness or sample. This indicates that the relative number of micro-inclusions in the vicinity of grain boundaries is, on average, relatively stable and not significantly impacted by the chosen gain boundary thickness. We do not know the inclination of the grain boundaries below the surface, which supports the use of a 300μm thick grain boundary which allows an inclination of the grain boundaries of to up to 33.3°."

[Figure]

*Figure 1 Grain size distribution within the 9 cm long physical properties samples. The indicated depths refer to the middle of the sample. Note the varying counts on the y-axis.*

**Ref2:** P8 l. 2: you are mentioning a "vertical girdle", but for me, on fig 1, the girdle is not vertical but rather inclined. Please correct or specify why you will consider this girdle to be "vertical".
Here we may have a misunderstanding:
**Reply:** As mentioned in the caption C-axis orientations of each section are projected into a horizontal plane as it is (good or not) custom in glaciology. Thus, their projection plane is not the plane of the microstructure map (Fig. b-k) but 90° to it. "Vertical" refers to the CPO in 3D and in relation to the flow direction of the ice stream. The visible inclination is due to the correction of the ice core/sample rotation towards the most likely position as explained in Westhoff et al., 2020.

**Ref2:** P8 l. 185-186: is it realistic to hope having a clear trend with depth based on only 10 samples with 9 coming from the same period and only 1 coming from another climatic period?
**Reply:** We only mention that there is no clear trend, not that we hoped to detect one. It is of course always nicer to have more samples, but since studies investigating the location of micro-inclusions are rare we think this is a valuable data set.

**Ref2:** Fig 3: I don't think the first sentence of the legend should appear here since there is no sign in the figure of stress or strain evaluation, only microstructure observations (interesting indeed!), so it is interpretation that should be given in the text.
**Reply:** We agree and changed the caption to: "Microstructural observations from all depths".

**Ref2:** I don't think either that these observations can explicitly tell that there are nucleated grains. You only observe small grains at GBs and triple junctions that you choose to interpret as nucleated grains, and that needs being justified. So maybe again, referring to nucleated grains should be let for the discussion part.
**Reply:** We agree and changed the caption to: "different types of subgrain boundaries (A-K), and small grains at grain boundaries".

**Ref2:** P10 l. 202: could you please give references about this hypothesis of "amorphous zone"?
**Reply:** This is indeed confusion wording, we changed it to "Our results are basically interfaces, with a discontinuity in lattice orientation accompanied by a zone of disorder of only a few molecule layers (Hobbs 1974, Petrenko and Whitworth, 1999).

**Ref2:** Part 3.3: mentioning a percentage of inclusion AT grain boundaries is not equivalent to mentioning them as being "in the vicinity of" or "close to" grain boundaries...
**Reply:** See above, we changed it to "in the vicinity of grain boundaries".

**Ref2:** In this part again, you observe 33 % of all inclusions to be located in your defined vicinity of GB, and this area (the vicinity of GB) represents 26% of the grain area... Can't you conclude out of that that the spread in homogeneous?
**Reply:** These values refer to the mean values over all samples, the individual samples show a higher variability. We do conclude that both parameters are similar and thus somehow homogeneous.
We clarified this in the discussion section: "The ratio $R_{GB}$ oscillates around 1, which describes, on average, a coherent distribution and a homogeneous spread of micro-inclusions in relation to grain boundaries throughout the core. $R_{GB}$ tends slightly towards larger values indicating that micro-inclusions are slightly more often located in the vicinity of grain boundaries than presumed from the grain boundary area. The derived p-values (Table 1, p-value) support these findings, but also emphasise the high variability between samples.".
We furthermore found a typo, the correct mean $R_{GB}$ of all samples is 1.19, not 1.29.

**Ref2:** What would those percentages become if you were to increase or decrease this vicinity area? What would your test of null significance become in these cases?
**Reply:** We did more analyses with grain boundary thicknesses of 100 and 200 μm. The results can be found in Table 3 (see above) and show that the ratio is rather constant and thus almost independent of the chosen grain boundary vicinity area. P-values change due to the smaller amount area/inclusions but constant total amount of inclusion per sample.
We think that the ground boundary thickness of 300 um is a good choice for an upper-limit assumption (see above), increasing it further would lead to untrustworthy result. This is especially true for samples with small grains where samples would be almost completely covered with grain boundaries.

**Discussion:**
**Ref2:** Please note that in many places in the text, and in the discussion, the reference to the companion paper is given. A large part of the discussion would not exist without the results of this companion paper, maybe that means that the two papers should be gathered together, since the present one is too weak to exist on its own.
**Reply:** We originally submitted the paper as one paper. The editor suggested to split it up into two parts. We apologise for any confusion regarding the term "companion paper". Both manuscripts were submitted at the same time, we were thus not able to add a proper citation to the companion paper. Both are clearly linked in TCD and we hoped that this connection would be done by TDC. We are optimistic that the newly added data (grain size distribution and grain boundary area analysis with 100 and 200 microns) fully supports the choice of having part 1 and part 2.
We disagree with the statement that a large part of the discussion would not exist without the companion paper. Sections 4.1, 4.2, and 4.4 only discuss the results of this study. Only l.344-356 and partly l. 376-393 of 4.3 discuss results of the companion paper, because it makes sense to discuss the interplay between localization and mineralogy in this part instead of part 2. We actually think it´s crucial to link both papers and see the combination of mineralogy and location as one of the strongest points of our work and as a gain towards a better understanding of the interplay between impurities and deformation.

**Ref2:** Another "sign" of that is that very few of the results presented here are mentioned in the discussion, where it is very often referred to other studies, or other techniques to use in order to "fill" the discussion, in particular when talking about the impact on deformation, or in the recrystallization part.
**Reply:** As mentioned above we disagree with this statement. We here show some examples from the deformation/recrystallization part of the discussion:
section 4.3.
p. 16 l. 324ff: "Even though roughly one third of all mapped micro-inclusions were located in proximity (300 μm upper limit assumption) to grain boundaries (1), we did not observe any direct indications of micro-inclusions involved in Zener pinning or drag. Direct indications would be e.g., a sharp edge at a second phase particle in an otherwise curved grain boundary segment or a bulging grain boundary restricted by a second phase particle (e.g., Fig. 2 in Stoll et al. (2021)) (Passchier and Trouw, 2005). „
p. 16 l. 333ff: "It is possible that EGRIP Holocene dust concentrations are below this vague threshold to impact grain size development significantly via pining, but unlikely because we also analysed samples comparable to Glacial ice, i.e. with high dust concentrations (discussed in a companion paper) and small grains (e.g., Fig. 1J and K)."

p. 17 l.337: "High impurity layers showed no microstructural evidence for grain boundary sliding, such as linked-up grain boundaries or rectangular and lath-shaped grains (Fig. 6 in Goldsby and Kohlstedt (1997), Fig. 2B in Kuiper et al. (2020b)). Our results (Fig. 3) indicate that solid micro-inclusions are not a main driver of e.g., grain size change via localised deformation along grain boundaries."

p. 17 l. 344ff: "The localisation of micro-inclusions at grain boundaries in our samples seems to be somewhat related to their mineralogy (Table 2, Fig. 8 in companion paper). In the case of solid micro-inclusions the interface properties between the ice and the particle play a crucial role and are thus probably influenced by the mineralogy and thus the surface properties of the inclusion at a grain boundary. We know from anti-freeze proteins (e.g., Bayer-Giraldi et al., 2018) that "ice-binding" properties result from similar surface structures (on the crystal lattice scale) of the second phase particle and the ice crystal. Less than one third of all feldspar, gypsum, and mica inclusions were located at grain boundaries, which is in good agreement with our average derived percentage of micro-inclusions at grain boundaries."

p. 17. L. 357: "While Bohleber et al. (2020) found element-specific localisation trends for dissolved impurities we show that it is more complex for solid micro-inclusions."

p. 18 l. 373: "That solid inclusions, in an almost constant temperature regime, do not tend to affix to grain boundaries, is indicated by our finding that the relative amount of mineral particles found at grain boundaries does not change significantly with depth (Table 1)."

p. 18 l. 382ff: "The post depositional movement of sulphate ions suggested by Barnes et al. (2003); de Angelis et al. (2013) could explain the high frequency of clusters (Fig. 4. In deep Vostok ice de Angelis et al. (2005) observed the relocation of mineral particles in large clusters related to grain boundary migration during extreme grain growth conditions. Our samples are from a different temperature regime and not from bottom ice and processes, such as abnormal grain growth, thus do not take place, but the overall mechanisms could be similar and should be investigated further."

p. 18 l. 392: "The observation that micro-inclusions in our samples, especially sulphates, form clusters indicates that these processes have to be considered in EGRIP ice as well."

section 4.4.
p. 19 l. 402f: "Indications of recrystallization processes, such as abundant subgrain boundaries, grain islands and bulging grain boundaries, were observed at all depths (Fig. 3) and indicate dynamic recrystallisation during deformation (in contrast to static recrystallization occurring without or after deformation) (Passchier and Trouw, 2005). Subgrain boundaries are stable structures of localised distortions of 405 the lattice (Faria et al., 2014b)."

p. 19 l. 408ff: "All three subgrain boundary types (normal, parallel, zigzag) described by Weikusat et al. (2009) were observed at all depths (Fig. 3) indicating a high mechanical anisotropy. Electron Backscatter Diffraction or X-ray Laue diffraction would be needed to identify possible active slip systems of the dislocations in our samples (Piazolo et al., 2008; Weikusat et al., 2011, 2017b; Chauve et al., 2017). In analogy to previous results from polar ice (NEEM, EDML) the high abundance of subgrain boundaries parallel to the basal plane could be basal twist boundaries, but also have the potential that internal stresses are sufficient to also produce non-basal dislocations."

p. 19 l. 414: "At the same depth, i.e. under similar boundary conditions, fewer fine grains undergo subgrain boundary formation than larger grains."

p. 19 l. 428ff: "The observed microstructure (Fig. 3) indicates dynamic recrystallization and thus active deformation by dislocation activity throughout the upper 1340 m of the EGRIP ice core. Active deformation probably continues with depth due to similar deformation mechanisms taking place below 1340 m as indicated by vertical girdle CPOs (Westhoff et al., 2020). Dynamic recrystallization does not act only at specific depth regimes, but throughout the ice column as was observed in other deep ice cores from less dynamic sites (e.g., Durand et al., 2008; Weikusat et al., 2009; Faria et al., 2014b)."

However, we further strengthened the discussion by discussing the grain size distribution data and the impact of the chosen grain boundary thickness as explained above.

**Ref2:** For illustration:
p 14 l. 282-283: why aren't these observations focused on cloudy bands not shown here?
**Reply:** We thought about it, but introducing visual stratigraphy data would go beyond the scope of this study. The data referred to is published on PANGAEA:
Weikusat, Ilka; Westhoff, Julien; Kipfstuhl, Sepp; Jansen, Daniela (2020): Visual stratigraphy of the EastGRIP ice core (14 m - 2021 m depth, drilling period 2017-2019). *PANGAEA*, https://doi.org/10.1594/PANGAEA.925014
Following your thoughts we clearly suggest a detailed study on cloudy bands on p. 14 l. 282: "Investigating those bands in detail from a microstructural and impurity perspective would help to better understand the relationship between impurity content, grain size and deformation."
To clarify this we add the citation to the text: "Visual stratigraphy at this depth (Weikusat et al., 2020) already shows that cloudy band,…".

**Ref2:** P 17 up to the end of part 4.3: everything that is discussed here is based on the results of the companion paper... And very few (if not nothing) on the presented results.
**Reply:** See answer above. We present the results in section 3.4 and do refer to the mineralogy of the inclusions and their impact, because it makes sense to have this discussion in this part. However, we also discuss the localization of inclusions in general (clusters etc.), which is described in this paper.

**Ref2:** Part 4.4: in this part, it is mostly mentioned the recrystallization markers observed, with very few links with the impurity observations, although the paper does not deal with recrystallization per se, and that all these observations have already been commented in previous studies. And as mentioned, there is not enough observations here to discuss the specificity of recrystallization mechanisms.
**Reply:** This paper investigates the location of inclusions in the microstructure, hints for recrystallisation are thus a logical result of this analysis. Combining different areas and approaches is one of the main ideas of both papers, because deciphering the ice flow history without the climatic impacts by impurity load is not feasible as the controversy by many studies on the topic has shown (see Stoll et al., 2021). As recrystallisation is the process group to determine grain size and shape, with a direct influence on rheology, we think our observations of recrystallisation should be included here. These observations have not been made for the EGRIP ice core yet and thus deserve to be published. We added some observations and thoughts on recrystallisation of EGRIP vs. NEEM ice, e.g., that dynamic recrystallisation in the upper hundred meters is stronger at EGRIP.

**Ref2:** p. 14 l. 268: NO, your result do not show that micro-inclusions are preferentially located AT grain boundaries. They eventually show that they, in some cases, are located nearby GBs... it is not the same at all! Especially when discussing GB pinning, influence of impurities on GB mobility, interaction with impurity interfaces, etc.
**Reply:** In the methods we explicitly define "at grain boundaries" as "Micro-inclusions located in the 300 μm large grain boundary area are classified as "at the grain boundary" and serve as upper-limit assumptions.". As explained above we changed the wording to be more precise and to avoid confusion.

**Ref2:** P 16 l. 329: do you think you have enough observations to be able to reject the hypothesis of Zener pinning?
**Reply:** No, which is never said in the text. But our observations agree with previous studies i.e. no indications of Zener pinning are observed. Thus, Zener pinning seems not to happen in our samples. However, we cannot and do not make assumptions about the role of insoluble impurities regarding Zener pining and therefore the hypothesis of Zener pinning in general.

**Ref2:** P17 l. 335-336: you analysed only one sample from Glacial ice, it is enough to give this statement?
**Reply:** We say that it is unlikely based on the samples from or similar to glacial conditions. We do not rule it out, which would indeed be impossible without a large amount of samples from the glacial. To clarify this we added the following sentence: "However, to confirm this assumption, a larger number of samples from the glacial period must be analysed."

**Conclusion:**

**Ref2:** "First systematic analysis": this sentence seems to me a little "two much" for 10 samples, and only one from the Late Glacial...
**Reply:** We disagree because this refers to the methodological aspect of investigating samples in 1) a high-resolution throughout the upper 1340 m of the core with
2) a range of data sets typically not intercompared even within the ice core community and
3) we are aiming for an overview if there is a development with depth/time.

Previous studies of this truly interdisciplinary approach (physics + chemistry) focus mainly on specific depths/locations in natural ice (e.g., Eichler et al., 2019) (). We do not claim to have done this on the entire core, which would be great but is unfortunately not possible with the time-demanding combination and measurement procedure (pre-preparation, pre-mapping, see methods) (see explanation above), and may never become possible.

**Ref2:** "Grain boundary are slightly preferred locations ...": I think that your results do not show that.
**Reply:** Our results show that for the assumed grain boundary width of 300 microns. At some samples grain boundaries are slightly preferred compared to the "available grain boundary area".
The statement indeed evokes the wrong conclusions, we thus changed the text to:
"We quantify the relationship between micro-inclusions and grain boundaries and show that roughly two-thirds of all micro-inclusions are located in the grain interior. However, at the half of all depths, grain boundaries are slightly more preferred locations than presumed from the grain boundary area."

**Ref2:** "The combination of optical microscopy and Cryo-Raman ...": this is not done in this paper.
**Reply:** The measurements were performed consecutively and Raman spectroscopy was used to verify that the observed dark dots are micro-inclusions as written in the methods (l.129): "These maps also provide the basis for a structured Raman spectroscopy study proving that the mapped dots are chemical impurities, i.e. micro-inclusions (Stoll et al., 2021b).".
While the majority of mineralogy-related results are presented in part 2 we present and evaluate the mineralogy of micro-inclusion close to grain boundaries in this paper. Combining these different approaches is a strength of this study. We thus leave the sentence as it is.

**Ref2:** "Observed clustering of micro-inclusions ...": do you observe enough of them?
**Reply:** We observed clustering in almost all samples throughout the core, which supports this statement. It is a very theoretical question if one has "enough observations". Since we are not aware of any publication discussing a quantitative way to deal with the clustering of micro-inclusions in ice the question of "enough" is difficult to answer completely. We are confident that our observations are enough in this regard.

**Ref2:** "Grain boundary mobility ...": maybe I missed something but I think that nothing in your results allows to assess the influence of the impurities you observe on this mobility. And this is what is said in the discussion.
**Reply:** We understand that you are missing a direct proof for this statement, but unfortunately this is not possible with the methods we had available. However, as in many other aspects we can and have to argue indirectly as the phenomenon (statistically difficult, but meso scale averaging) is clear: "dirty ice – small grains". We discuss our interpretation with respect to grain size evolution and its theory in quite some detail, e.g:

p. 16 l. 322ff „Zener pinning and drag of grain boundaries by impurities are suggested to be major mechanisms impacting grain size (e.g., Smith, 1948; Alley et al., 1986a; Fisher and Koerner, 1986; Alley and Woods, 1996; Paterson, 1991; Weiss et al., 2002; Durand et al., 2006). Even though roughly one third of all mapped micro-inclusions were located in proximity (300 μm upper limit assumption) to grain boundaries (1), we did not observe any direct indications of micro-inclusions involved in Zener pinning or drag. Direct indications would be e.g., a sharp edge at a second phase particle in an otherwise curved grain boundary segment or a bulging grain boundary restricted by a second phase particle (e.g., Fig. 2 in Stoll et al. (2021)) (Passchier and Trouw, 2005)."

p.17 l.364ff "The density anomaly of water, in contrast to metals or minerals, leads to molecules in grain boundaries being packed more closely than in the lattice and amorphous water veins (liquid-like (e.g., Mader, 1992) or solid-like) thus exist at grain boundaries (Azuma et al., 2012). Depending on the conditions, mainly changed temperature, but possibly also pressure and salt content, these veins can become liquid-like and thus allow premelting and "slippery" grain boundaries as suspected by Kuiper et al. (2020a). Comparably to air bubbles during normal grain growth, dissolved impurities are mobile enough to interact and affix on migrating

grain boundaries (Azuma et al., 2012), thus influencing their migration mobility. The concentration of dissolved impurities at grain boundaries could thus increase with time resulting eventually in a decreasing grain boundary mobility."

p. 17 l373 "That solid inclusions, in an almost constant temperature regime, do not tend to affix to grain boundaries, is indicated by our finding that the relative amount of mineral particles found at grain boundaries does not change significantly with depth (Table 1)."

**References:**
Watanabe, O., Jouzel, J., Johnsen, S., Parrenin, F., Shoji, H., and Yoshida, N.: Homogeneous climate variability across East Antarctica over the past three glacial cycles, Nature, 422, 509–512, https://doi.org/10.1038/nature01525, 2003.
Treverrow, A., Budd, W. F., Jacka, T. H., & Warner, R. C. (2012). The tertiary creep of polycrystalline ice: Experimental evidence for stress-dependent levels of strain-rate enhancement. *Journal of Glaciology*, *58*(208), 301–314. https://doi.org/10.3189/2012JoG11J149

---

## Author Response (AR2)

**Comments to the editor**:
Dear Dr. Sandells,

Many thanks for this update and the positive outcome of our revisions. We are happy to address the mentioned corrections below.
Thanks again for a smooth and fair review process, we appreciate it.

With best wishes,
Nico

**Line 156, 157. Change R>1 to R_GB>1, similarly for R<1**
Both cases changed to $R_{GB}$ (l. 156 +157).

**Line 211-222. Join into one paragraph. 'From now on, we will always refer to a grain boundary thickness of 300 µm' - please repeat justification here e.g. 'for consistency with Eichler et al., (2017)'**
The addressed lines are now one single paragraph (l. 211-223). The sentence was changed to "From now on, we will always refer to a grain boundary thickness of 300 µm for consistency with Eichler et al. (2017)." (l. 219-220).

**Line 237. State null hypothesis (which I think is that impurities are no more likely to be found in the vicinity of the boundary). Line 238 is ambiguous and seems to ignore the case where R_GB<1 and p<0.05 (at 514.44 m depth).**
We state the null hypothesis as follows:" $A_{GB}$ and $I_{GB}$ were used to perform null hypothesis significance testing for ten samples with an alpha of 0.05 as the cutoff for significance. The null hypothesis states that micro-inclusions are preferentially located in the vicinity of grain boundaries compared to the grain interior. If the p-value is smaller than 0.05, we reject this null hypothesis." (l. 238-240).
We now specifically mention the case at 514 m of depth: "A small p-value often correlates with a high RGB (except at 514.44 m) while a high p-value usually correlates with a low $R_{GB}$." (l. 243-244).

**Figure 7 caption. 'less inclusions' -> 'fewer inclusions' (sorry I did not catch this before)**
Changed to "fewer inclusions" in Figure 7 caption.

**Section 4.2.2. Refer to 'number of inclusions' or 'proportion of inclusions' rather than 'amount' and in that case the text should probably read the 'majority are' rather than 'majority is'.**
We changed "amount of inclusions" to "number of inclusions" and "majority is" to "majority are" throughout the text.

**Line 317. 'chosen gain boundary' -> 'chosen grain boundary'**
Changed to "grain boundary" (l. 316).
**Line 372, 377: 'pining' -> 'pinning'**
Changed to "pinning" (.l 373+ 376).